# Stochastic Optimization in Semi-Discrete Optimal Transport: Convergence Analysis and Minimax Rate

**Ferdinand Genans**[1*]    **Antoine Godichon-Baggioni**[1]
**François-Xavier Vialard**[2]    **Olivier Wintenberger**[1,3]
Sorbonne Université, CNRS, LPSM[1]
Université Gustave Eiffel, CNRS, LIGM[2]
Wolfgang Pauli Institute[3]
{ferdinand.genans-boiteux, antoine.godichon_baggioni,
olivier.wintenberger}@sorbonne-universite.fr
francois-xavier.vialard@u-pem.fr

## Abstract

We investigate the semi-discrete Optimal Transport (OT) problem, where a continuous source measure $\mu$ is transported to a discrete target measure $\nu$, with particular attention to the OT map approximation. In this setting, Stochastic Gradient Descent (SGD) based solvers have demonstrated strong empirical performance in recent machine learning applications, yet their theoretical guarantee to approximate the OT map is an open question. In this work, we answer it positively by providing both computational and statistical convergence guarantees of SGD. Specifically, we show that SGD methods can estimate the OT map with a minimax convergence rate of $\mathcal{O}(1/\sqrt{n})$, where $n$ is the number of samples drawn from $\mu$. To establish this result, we study the averaged projected SGD algorithm, and identify a suitable projection set that contains a minimizer of the objective, even when the source measure is not compactly supported. Our analysis holds under mild assumptions on the source measure and applies to MTW cost functions, which include $\|\cdot\|^p$ for $p \in (1, \infty)$. We finally provide numerical evidence for our theoretical results.

## 1   Introduction

Optimal Transport (OT) has become a central tool in machine learning for comparing and manipulating probability measures. A particularly important variant is the *semi-discrete* setting, where a continuous source distribution $\mu$ is transported to a discrete target measure $\nu$. This formulation arises naturally in a wide range of applications, including image processing [16, 22], statistics [11, 18], and generative modeling [4, 9, 24]. Its hybrid structure bridges the gap between fully continuous and fully discrete formulations, allowing for expressive modeling while remaining more amenable to scalable numerical methods.

Despite its practical appeal, solving semi-discrete OT efficiently and reliably at scale remains challenging. Existing methods with convergence guarantees often require *full knowledge* of the source density and are typically confined to *low-dimensional settings*. For instance, Newton-type methods [27, 23, 21] and combinatorial [2, 1] methods have been developed and are provided with convergence rates guarantees. However, these techniques become impractical in high-dimensional settings, since they need full knowledge of the source measure, and employ constructions that suffer from the curse of dimensionality such as meshes representation of the source measure.

In high dimensional and/or when the source measure can only be accessed through samples, SGD and its variants have become a popular choice for solving semi-discrete OT, especially in applications such as generative modeling [4, 9, 24]. These methods solve the semi-dual formulation of the OT

39th Conference on Neural Information Processing Systems (NeurIPS 2025).

problem using only i.i.d. samples from $\mu$ and scale well to large data, not requiring to store samples. Yet, *a fundamental theoretical gap remains*: while SGD methods are widely used in practice, they lack convergence guarantees for approximating the OT map.

Providing convergence rates for SGD to approximate the semi-discrete OT map is both a computational and a statistical problem, since using more gradient steps in SGD for the semi-discrete setting is equivalent to using more samples. From a statistical point of view, recent work by Pooladian et al. [31] offers promising evidence for the convergence of SGD: they show that semi-discrete OT escapes the curse of dimensionality, unlike the continuous setting [15], and that a convergence rate of $\mathcal{O}(1/\sqrt{n})$ with $n$ samples is achievable when the cost to move mass is the quadratic cost $\frac{1}{2}\|\cdot\|^2$ and the source measure is compactly supported. Moreover, this rate is minimax optimal for estimating the OT map. While their approach requires solving a discrete OT problem, i.e., first sampling points from $\mu$ and then solving the corresponding empirical problem, their key result motivates studying SGD as a way to accurately estimate the OT map in an online setting, where the estimator is refined as more samples become available, without needing to store them, as is often required in practice, for instance in generative modeling tasks

Taken together, these observations highlight two major open questions that are answered positively in this work:
(i) Can we establish convergence guarantees for SGD-based algorithms in the semi-discrete OT setting for OT map estimation, especially when only samples from $\mu$ are available?
(ii) Can we obtain statistical guarantees for the estimation of OT quantities (e.g., cost, potential, map) beyond the compact and quadratic setting of Pooladian et al. [31]?

**Contributions.** We establish convergence guarantees for SGD-based algorithms applied to the non-regularized semi-dual formulation of optimal transport. We focus our analysis on the convergence of the averaged Projected Stochastic Gradient Descent (PSGD) algorithm, relying on a key result: the existence of a compact projection set $\mathcal{C}$ (Lemma 3.1) that contains a minimizer of the semi-dual objective, even when the source measure $\mu$ has unbounded support. To the best of our knowledge, this is the first time such a projection set has been identified in the semi-discrete OT setting without assuming boundedness of $\mu$. This projection set allows us to derive key properties of the semi-dual functional, including a global weak form of strong convexity on $\mathcal{C}$ in all our settings (Lemma 4.2), and to extend second-order regularity results that were previously known only in the compact setting (Prop 4.1) , such as local strong convexity near the optimum.

The convergence rates for OT quantities, obtained when $n$ samples are drawn from $\mu$ (equivalently, when PSGD is run for $n$ iterations), answer the two questions posed in the introduction. These results are summarized in Table 1. Moreover, we show our convergence rate to the OT map is minimax optimal (Theorem 5.3), giving the first statistical guarantees in semi-discrete OT for MTW (defined in Section 2.1) and quadratic costs on unbounded support.

Table 1: Summary of the convergence rate achieved by PSGD for the estimation of OT quantities

| Costs | OT cost | OT potential | OT map | Non compact |
|---|---|---|---|---|
| MTW costs | $\mathcal{O}(1/n)$ [Cor 4.6] | $\mathcal{O}(1/n)$ [Th 4.5] | $\mathcal{O}(1/\sqrt{n})$ [Cor 5.2] | No |
| Quadratic | $\mathcal{O}(1/n)$ [Cor 4.6] | $\mathcal{O}(1/n)$ [Th 4.5] | $\mathcal{O}(1/\sqrt{n})$ [Cor 5.2] | Yes |
| All other costs | $\mathcal{O}(1/\sqrt{n})$ [Th 3.2] | No guarantees | No guarantees[1] | Yes |

## 2 Background

### 2.1 Optimal Transport

Considering a source and target probability measures $\mu \in \mathcal{P}(\mathbb{R}^d)$ and $\nu \in \mathcal{P}(\mathbb{R}^d)$, with a cost $c : \mathbb{R}^d \times \mathbb{R}^d \to \mathbb{R}^+$ to transfer mass, the OT problem is defined as:

$$\mathrm{OT}_c(\mu, \nu) := \min_{\pi \in \Pi(\mu, \nu)} \int_{\mathbb{R}^d \times \mathbb{R}^d} c(x, y) \mathrm{d}\pi(x, y), \tag{1}$$

---

[1]Not even necessarily defined.

where $\Pi(\mu,\nu) := \left\{ \pi \in \mathcal{P}(\mathbb{R}^d \times \mathbb{R}^d); \ \pi(\cdot \times \mathbb{R}^d) = \mu(\cdot), \ \pi(\mathbb{R}^d \times \cdot) = \nu(\cdot) \right\}$ is the set of joint probability measures on $\mathbb{R}^d \times \mathbb{R}^d$ with marginals $\mu$ and $\nu$.

In this article, we assume that $\mu$ is continuous, while the target measure takes the form $\nu = \sum_{i=1}^{M} w_i \delta_{\{y_i\}}$, where $\mathbf{w} = (w_1, ..., w_M)$ are its probability weights and $(y_1, ..., y_M)$ its support. We are mainly interested in OT problems where (1) has a unique deterministic solution, which we refer to as the OT map. This holds under mild assumptions in our semi-discrete setting, as guaranteed by the generalized Brenier's Theorem.

**(Generalized) Brenier's Theorem** ([34], Th. 10.28). Suppose that $\mu$ is an absolutely continuous probability measure, and $\nu$ is discrete. Then, if the cost $c$ satisfies the MTW properties, as when $c = \|\cdot\|^p$ with $p \in (1, \infty)$, and if the OT cost in (1) is finite then it admits, up to negligible sets, a unique solution of the form $\gamma(dx, dy) = \mu(dx)\, \delta_{T_{\mu,\nu}(x)}(dy)$, $\forall (x, y)$, where:

$$T_{\mu,\nu}(x) = x - \nabla f^*(x).$$

$T_{\mu,\nu}$ is referred to as the **OT map**. Moreover, $f^*$ is a solution of the dual problem of (1), given by:

$$\mathrm{OT}_c(\mu, \nu) = \max_{f(x)+g(y) \,\leq\, c(x,y)} \int_{\mathbb{R}^d} f(x)\mathrm{d}\mu(x) + \int_{\mathbb{R}^d} g(y)\mathrm{d}\nu(y). \tag{2}$$

**The semi-dual problem.** The semi-dual formulation of (2) is particularly useful in the semi-discrete setting and can be expressed as a concave finite-dimensional problem:

$$\mathrm{OT}_c(\mu, \nu) = \max_{\mathbf{g} \in \mathbb{R}^M} \left( H(\mathbf{g}) := \int_{\mathbb{R}^d} \mathbf{g}^c(x)\mathrm{d}\mu(x) + \sum_{i=1}^{M} w_j g_j \right), \tag{3}$$

where $\mathbf{g} = (g_1, ..., g_M)$, and for all $x \in \mathbb{R}^d$, the (vectorial) $c$-transform is defined as $\mathbf{g}^c(x) := \min_{i \in [\![1,M]\!]} \{ c(x, y_i) - g_i \}$. For any $\mathbf{g}$, the $c$-transform also defines the Laguerre cells $\mathbb{L}_j^c(\mathbf{g})$ for $j \in [\![1, M]\!]$ as

$$\mathbb{L}_j^c(\mathbf{g}) := \{ x \in \mathcal{X} \mid \mathbf{g}^c(x) = c(x, y_j) - g_j \}.$$

Using this formulation and under Brenier's generalized theorem, the OT map can be described as $T_{\mu,\nu}(x) = x - \nabla (\mathbf{g}^*)^c(x) = x - y_i$ for $x$ inside $\mathbb{L}_i^c(\mathbf{g}^*)$, where $\mathbf{g}^*$, that we refer to as the discrete **OT potential** solves (3).

**MTW costs.** For costs satisfying the Ma-Trudinger-Wang properties, referred to as MTW costs, such as $\|\cdot\|^p, p \in (1, \infty)$ or Bregman divergences $\phi(x) - \phi(y) - \langle \nabla \phi(y), x - y \rangle$ for strictly convex $\phi$, we observe improved properties for the function $H$ in (3). Specifically, the differential is defined almost everywhere, except on a $\mu$-negligible set, and [21] proved that $H$ is even locally smooth and strongly convex on the orthogonal complement of $\mathbf{1}$. Moreover, all these costs satisfy Brenier's generalized theorem. For a detailed definition of such costs in the semi-discrete setting, see [21] or Appendix A.

## 2.2 Stochastic approach for the semi-dual problem.

In the semi-discrete setting, the semi-dual formulation is particularly appealing because, even when $\mu$ is a continuous measure, it reduces to a finite-dimensional problem. Efficient (quasi)-Newton schemes exist to solve this problem in low-dimensional settings when the density of $\mu$ is known [27, 23, 21]. In the more general scenario, where the dimension can be high and/or only sampled points from $\mu$ are available, we reformulate the semi-dual OT problem as a convex expected minimization problem, defined as:

$$\min\{H(\mathbf{g}) := \mathbb{E}_{X \sim \mu}[h(\mathbf{g}, X)] \mid \mathbf{g} \in \mathbb{R}^M\}, \tag{4}$$

where for all $\mathbf{g} \in \mathbb{R}^M, x \in \mathbb{R}^d$, $h(\mathbf{g}, x) = -\mathbf{g}^c(x) - \sum_{j=1}^{M} w_j g_j$. Note that we deliberately multiplied the semi-dual functional by $-1$ to frame it as a convex minimization problem instead of a concave maximization problem; however, this is not a universal convention in the literature.

No matter the cost, $H$ is subdifferentiable everywhere, and we consider the subdifferential $\partial H(\mathbf{g}) = \mathbb{E}_{X \sim \mu}[\partial_{\mathbf{g}} h(\mathbf{g}, X)]$, where for $x \in \mathbb{R}^d$ and $j \in [\![1, M]\!]$, we define:

$$\partial_{\mathbf{g}} h(\mathbf{g}, x)_j = \mathbb{1}_{x \in \mathbb{L}_j(\mathbf{g})} - w_j.$$

As long as $x$ is in the interior of a Laguerre cell, this subdifferential $\partial H$ is, in fact, a differential that we note $\nabla H$. Moreover, $\partial_{\mathbf{g}} h(\mathbf{g}, X)$ is an unbiased estimator of $\partial H(\mathbf{g})$. Given access to samples from $\mu$ naturally leads to the study of stochastic gradient descent schemes of the form

$$\mathbf{g}_n = \mathbf{g}_{n-1} - \gamma_n \partial_{\mathbf{g}} h(\mathbf{g}, X_n) \,,$$

where we start from an initial point $\mathbf{g}_0 \in \mathbb{R}^M$, and at each iteration $n$, draw a sample $X_n$ and take a gradient step with step size $\gamma_n > 0$ (also referred to as the learning rate).

SGD algorithms are well-suited for this setting [4, 9, 24], as they adapt to the number of samples drawn, have linear $\mathcal{O}(M)$ complexity per iteration, efficiently handle mini-batches through GPU parallelization, and do not require storing the drawn samples. Directly solving (4) also helps avoid discretization bias when estimating OT quantities [8, 17], which can be crucial in some applications of semi-discrete OT [9]. However, the specific structure of the OT semi-dual problem makes analyzing the convergence of SGD algorithms particularly challenging, especially regarding convergence to the optimizer $\mathbf{g}^*$. Unlike standard cases, it does not fall within the class of well-behaved problems, such as those that are globally strongly convex.

**Regularization of the semi-dual.** The idea of formulating the semi-dual OT problem as the minimization of an expectation and avoiding discretization by using SGD algorithms was introduced in [17]. However, possibly due to the lack of globally favorable properties of $H$, they propose using the entropy-regularized version of $H$, referred to as the entropic semi-dual $H_\varepsilon$, where $\varepsilon$ is the regularization parameter. This results in a globally $1/\varepsilon$-smooth problem and as $\varepsilon$ vanishes, $H_\varepsilon$ converges to $H$. A broader class of regularizer was also introduced and studied in [33]. Unfortunately, the theoretical analysis of SGD algorithms for the regularized problem reveals prohibitive constants in $\varepsilon^{-1}$ and higher in [33] and even for the entropic regularizer [7], making the use of a small $\varepsilon$ impractical for theoretical guarantees. Thus, avoiding both regularization and discretization bias to solve the semi-discrete problem highlights the relevance of studying SGD for the non-regularized OT problem.

## 3 Projected Stochastic Gradient Descent on the Semi-Dual OT Problem

### 3.1 Localizing a projection set

In convex optimization, particularly in an online or stochastic setting, localizing a set to restrict the optimization domain and using a projection step in the gradient descent scheme can be very useful, permitting straight-forward convergence proofs [20]. Our first lemma addresses this idea in the context of the OT semi-dual problem, showing that even when the support of $\mu$ is not compact, it is still possible to localize a $\| \cdot \|_\infty$-ball within which a minimizer of the semi-dual function $H$ exists. This projection set is formally defined in Lemma 3.1.

**Lemma 3.1** (Existence of a projection set)**.** *Suppose that the OT problem is well-posed, strong duality holds and* (4) *admits a minimum. Then, there exists a minimizer* $\mathbf{g}^*$ *contained in the set*

$$\mathcal{C} := \left\{ \mathbf{g} \in \mathbb{R}^M \mid |g_j| \leq \|c\|_{K,\infty} \right\},$$

*where* $\|c\|_{K,\infty} := \sup_{x \in K, \, j \in [\![1,M]\!]} |c(x, y_j)|$, *for any compact $K$ satisfying $\mu(K) \geq 1 - \frac{1}{2} \min_j w_j$.*

While the existence of a $\| \cdot \|_\infty$-ball was previously established under the assumption that the cost function is uniformly bounded on $\mathbb{R}^d \times \mathbb{R}^d$, or when both measures have bounded support, we extend this result to the more general semi-discrete setting. On its own, this finding may enable further theoretical developments in semi-discrete OT, where a bounded potential is often required [3, 13].

**Example 1.** Consider $\mu$ as the standard Gaussian on $\mathbb{R}^3$, the cost function $c = \frac{1}{2}\|x - y\|^2$, and $\nu$ as a discrete measure with $10^7$ points in $[0, 1]^3$ and uniform weights. In this setting, it was previously believed that the Brenier potential could not be bounded, since $c$ is not bounded on $\mathbb{R}^3 \times [0, 1]^3$. However, using Lemma 3.1, we can take the ball $B(0, 6)$, which allows us to restrict our search for the potential within a $\| \cdot \|_\infty$-ball of radius 18.

Incorporating this projection step into our SGD scheme has several advantages: (i) it significantly enhances both the practical performance and theoretical convergence of the algorithm; and (ii)

the computational complexity of the projection step is $\mathcal{O}(M)$, as it simply involves clipping each coordinate of the vector. The projector is defined as

$$\text{Proj}_{\mathcal{C}}(\mathbf{g}) : \mathbf{g} \in \mathbb{R}^M \mapsto \arg\min\{\|\mathbf{g} - \mathbf{g}'\|; \mathbf{g}' \in \mathcal{C}\}.$$

Based on this projection step, we derive the PSGD algorithm to minimize $H$, as presented in Algorithm 1. Note that, in practice, this requires knowledge of a compact set $K$, which we assume to be either given or previously estimated. The estimation is for instance trivial when the source is the standard Gaussian as in many applications (e.g. generative modeling). When the density is unknown, we can still have high probability guarantees as presented in the next paragraph.

**Estimation of the projection set $K$.** When the density of $\mu$ is unknown and we only have access to samples, we can, for instance, estimate the projection set $K$ using a centered ball $B(0, R)$ such that, with high probability, $\mu(B(0, R)) \geq 1 - \frac{1}{4} w_{\min}$. This is a standard quantile estimation problem for the norm $\|X\|$, where $X \sim \mu$, and the empirical CDF $F_n(r) = \frac{1}{n} \sum_{i=1}^n \mathbf{1}_{\|X_i\| \leq r}$ can serve as an estimator for $R$. Indeed, we can take $R = \inf\left\{r : F_n(r) \geq 1 - \frac{1}{8} w_{\min}\right\}$. By the Dvoretzky-Kiefer-Wolfowitz inequality [26], for a sample size $n \geq \frac{32 \log(2/\delta)}{w_{\min}^2}$, it holds with a

---

**Algorithm 1** Projected Stochastic Gradient Descent (PSGD)

---

**Parameters:** $\gamma_1 > 0, b \in \left[\frac{1}{2}, 1\right)$
Initialize $\mathbf{g}_0 \in \mathcal{C}$ and $\overline{\mathbf{g}}_0 = \mathbf{g}_0$
**for** $k = 1$ to $n$ **do**
    Draw $x_k \sim \mu$
    $\mathbf{g}_k = \text{Proj}_{\mathcal{C}}\left(\mathbf{g}_{k-1} - \frac{\gamma_1}{k^b} \partial_{\mathbf{g}} h(\mathbf{g}_{k-1}, x_k)\right)$
    $\overline{\mathbf{g}}_k = \frac{1}{k+1} \mathbf{g}_k + \frac{k}{k+1} \overline{\mathbf{g}}_{k-1}$
**end for**
**return** $\mathbf{g}_n$ and $\overline{\mathbf{g}}_n$

---

probability of at least $1 - \delta$ that $\mu(B(0, R)) \geq 1 - \frac{1}{4} w_{\min}$. This provides a dimension-free sample-complexity bound for estimating $K$.

### 3.2 A first convergence of PSGD in the general setting

As a first consequence of our projection step, we can directly establish a convergence rate for PSGD on the OT cost in a general setting, without assuming additional regularity of the cost function or the source measure. The proof follows a classical result for PSGD algorithms, and our projection step provides insights into the choice of the learning rate $\gamma_1$. Additionally, it allows us to recover the rate $\mathcal{O}(1/\sqrt{n})$ from [17] for the averaged iterates of SGD when applied to the entropy-regularized semi-dual.

**Theorem 3.2** (PSGD in the general setting). *In the general setting, choosing the learning rate $\gamma_n = \gamma_1/n^b$ with $\gamma_1 = \text{Diam}(\mathcal{C})/2\sqrt{2}$ and $b = 1/2$, we obtain*

$$\mathbb{E}\left[H\left(\overline{\mathbf{g}}_n\right) - H\left(\mathbf{g}^*\right)\right] \leq \frac{4\sqrt{2}\,\text{Diam}(\mathcal{C})}{\sqrt{n}}.$$

Although most of our results focus on MTW costs, as discussed in the next section, we also establish a convergence rate in a more general setting. This broader setting lies outside the scope of the generalized Brenier theorem and an OT map may not exist or be unique. However, estimating the OT cost can still be useful in certain applications, such as when $c = \|\cdot\|$, corresponding to the 1-Wasserstein distance. Thanks to our projection step, the same convergence rate carries over to other SGD-based methods, such as Adagrad [14], which is well-suited when the projection set is a hypercube [28, Section 4.2.4].

## 4 Convergence analysis of PSGD and minimax estimation for MTW costs

We now focus on costs satisfying the MTW properties, with particular attention to the cost $c(x, y) = \frac{1}{2}\|x - y\|^2$, as it is the most commonly used. For this cost, we extend our results to include non-compactly supported source measures. This setting is used, for instance, in [4, 24], where a standard Gaussian is mapped to a discrete distribution. Our objective is to establish the convergence rate of PSGD in approximating the true OT map and cost by estimating the Brenier potential $\mathbf{g}^*$. We make the following assumptions, distinguishing between the compact case, where we treat all MTW costs, and the non-compact case, where we focus solely on the quadratic cost $\frac{1}{2}\|\cdot\|^2$.

**Assumption A** (Compact case). The cost $c$ satisfies the MTW condition, $\text{Supp}(\mu)$ is bounded and $c$-convex (see Appendix A), and $\mu$ satisfies a weighted $(1,1)$ Poincaré-Wirtinger inequality: there exists $C_{\text{pw}} > 0$ such that for all $f \in \mathcal{C}^1(\mathbb{R}^d)$,

$$\|f - \mathbb{E}_\mu[f]\|_{L^1(\mu)} \leq C_{\text{pw}}\|\nabla f\|_{L^1(\mu)}. \tag{PW}$$

Note that (PW) relates to the local strong convexity of the semi-dual problem near the optimum and is a common assumption as in [21, 6]. Moreover, in the compact setting, the assumption that the density $f_\mu$ is bounded from above and below by strictly positive values is a common assumption, as in [31]. This compact assumption implies (PW).

Regarding the non-compact case, our assumptions are satisfied notably, for non-degenerate Gaussians, finite mixtures of non-degenerate Gaussians, and heavy-tailed distributions such as Student distributions with degree of freedom larger than 2. While a broader class of MTW costs could potentially be covered, perhaps under stronger assumptions regarding the source measure, we deliberately omit these cases to avoid further technical complexity. Moreover, to the best of our knowledge, even for the quadratic cost, there are no existing theoretical results in semi-discrete OT when the support of $\mu$ is unbounded. We further provide in Appendix A an example where assumption (B3) fails and as a consequence, the Hessian is not defined.

**Assumption B** (Non-compact case).

**(B1)** The cost is quadratic: $c(x,y) = \frac{1}{2}\|x - y\|^2$ and the measure $\mu$ has a finite second-order moment.

**(B2)** There exists a compact set $K \subset \mathbb{R}^d$ with $\mu(K) \geq 1 - \frac{1}{4}w_{\min}$, such that the probability measure $\mu_K$ with density $f_{\mu_K}(x) := c_K f_\mu(x)\mathbf{1}_K(x)$ satisfies a (PW) inequality.

**(B3)** The density $f_\mu$ satisfies the following integrability and regularity condition: for $R > 1$ and $r \geq 1$, define
$$f_\mu^R := f_\mu \cdot \mathbf{1}_{\|x\| \leq R}, \quad f_\mu^{R+r} := f_\mu \cdot \mathbf{1}_{R+(r-2) \leq \|x\| \leq R+r}.$$
Assume there exist $R > 1$, $C > 0$ and a modulus of continuity $\omega$ such that for all $\delta > 0$,

$$\sum_{r=0}^\infty (R+r)^{d-1}\omega_{f_\mu^{R+r}}^{R+r}(\delta) \leq C\,\omega(\delta), \quad \sum_{r=0}^\infty (R+r)^{d-1}C_{f_\mu}^{R+r} < \infty, \tag{5}$$

where $C_{f_\mu}^{R+r} := \sup_{x \in \mathbb{R}^d} f_\mu^{R+r}(x)$ and $\omega_{f_\mu}^{R+r}$ is the modulus of continuity of $f_\mu^{R+r}$.

## 4.1 Properties of the semi-dual $H$

In our context, it is known that the discrete Brenier potential $\mathbf{g}^* \in \mathbb{R}^M$ is unique only up to a transformation of the form $\mathbf{g}^* + a\mathbb{1}_M$ with $a \in \mathbb{R}$. For clarity, we fix $\mathbf{g}^*$ to be the Brenier potential such that $\mathbf{g}^* \in \text{Vect}(\mathbb{1}_M)^\perp$. Without losing information, we thus restrict our analysis to the orthogonal complement of the subspace spanned by the vector $\mathbb{1}_M$, $\text{Vect}(\mathbb{1}_M)^\perp$. Therefore, for any $\mathbf{g}, \mathbf{g}' \in \mathbb{R}^M$, we define

$$\|\mathbf{g} - \mathbf{g}'\|_v = \|\text{Proj}_{\text{Vect}(\mathbb{1}_M)^\perp}(\mathbf{g} - \mathbf{g}')\|,$$
$$\langle \mathbf{g}, \mathbf{g}'\rangle_v = \langle \text{Proj}_{\text{Vect}(\mathbb{1}_M)^\perp}(\mathbf{g}), \text{Proj}_{\text{Vect}(\mathbb{1}_M)^\perp}(\mathbf{g}')\rangle.$$

We start by stating the second-order regularity of $H$ in our setting.

**Proposition 4.1.** *Under Assumption A or B, the function $H$ is differentiable everywhere on $\mathbb{R}^M$, and we denote its gradient by $\nabla H$. Moreover, there exists a radius $r > 0$ such that on the ball $B(\mathbf{g}^*, r)$, $H$ is $C^2$ and strongly convex on $B(\mathbf{g}^*, r)$. If, in addition, $f_\mu$ is $\alpha$-Hölder continuous with $\alpha \in (0,1]$, then the Hessian of $H$ is also $\alpha$-Hölder continuous.*

Naturally, the smallest eigenvalue of the Hessian is 0, with $\mathbb{1}_M$ as its eigenvector. However, we still refer to the strong convexity of $H$ since we focus on the orthogonal complement of $\text{Vect}(\mathbb{1})$. Notably, we extended the definition of the Hessian, originally provided in [21] for the compact case, to include the quadratic Euclidean cost in the non-compact setting. Furthermore, while the local strong convexity of $H$ was established in [21] for the compact case, it was defined over the set of vectors $\mathbf{g}$ such that the measures of all Laguerre cells are bounded by a positive constant. To formulate

this result with respect to a ball, we also required a result on the quantitative stability of the measures of Laguerre cells. Additional details on this quantitative stability will be provided in section 5.1.

As a corollary of these results, we derive the following lemma, which stems from our projection step and can be viewed as a weak form of strong convexity of $H$ on $\mathcal{C}$, also referred to as **Restricted Strong Convexity** (RSC) [37].

**Lemma 4.2.** *Under Assumptions A or B, there exists $\eta > 0$ such that $H$ satisfies a RSC property, uniformly for all $\mathbf{g} \in \mathcal{C}$:*

$$\langle \nabla H(\mathbf{g}), \mathbf{g} - \mathbf{g}^* \rangle_v \geq \eta \|\mathbf{g} - \mathbf{g}^*\|_v^2.$$

**Direct convergence guarantees under RSC.**   As a direct consequence of the RSC property on $\mathcal{C}$, Projected SGD achieves $\mathcal{O}(1/n)$ convergence for our OT problem when using the step size $\gamma_t = \frac{1}{\eta t}$. However, our analysis does not provide a precise estimate of the parameter $\eta$, which is required to effectively implement this learning rate in practice, and is particularly difficult to obtain in the OT setting. This limitation motivates the following section, where we study PSGD with a learning rate of the form $\gamma_t = \gamma_1/t^b$ for $b < 1$. This variant achieves optimal convergence rates without requiring prior knowledge of $\eta$.

*Remark* 4.3. Since RSC implies the Quadratic Growth (QG) condition $H(\mathbf{g}) - H(\mathbf{g}^*) \geq \eta\|\mathbf{g}-\mathbf{g}^*\|_v^2$, the same convergence guarantees also hold for other SGD variants that rely on either RSC or QG assumptions, such as S-Adagrad [10].

## 4.2   Convergence rate of PSGD

**Convergence of the non-averaged iterates**   Building on the RSC of $H$ from Lemma 4.2, we derive the convergence rate of the non-averaged iterates of PSGD, which mirrors the convergence behavior observed in the strongly convex setting.

**Theorem 4.4** (Non-averaged iterates)**.** *Under Assumptions A or B, and for any decay schedule of the form $\gamma_n = \gamma_1/n^b$ with $\gamma_1 > 0$ and $b \in (1/2, 1)$, we have the convergence rate*

$$\mathbb{E}[\|\mathbf{g}_n - \mathbf{g}^*\|_v^2] = \mathcal{O}\left(1/n^b\right).$$

As $b$ approaches 1, we observe a nearly $\mathcal{O}(1/n)$ rate for the OT potential. In the next section, we further show that this convergence rate is achievable by the averaged iterates sequence $\overline{\mathbf{g}}_n$ and that it is minimax optimal, highlighting the strong performance of PSGD.

**Convergence of the averaged iterates**   In convex stochastic optimization, averaging the iterates of the SGD scheme is a widely used technique, as it enables achieving an optimal $\mathcal{O}(1/n)$ convergence rate for strongly convex functions without requiring knowledge of the strong convexity parameter, and regardless of the decay $b \in (1/2, 1)$ for the gradient steps [30, 29]. Moreover, the averaged scheme can adapt to the local strong convexity of the objective function, even when global strong convexity does not hold [5].

Note that, as stated in Proposition 4.1, $H$ is locally strongly convex, and as stated after Lemma 4.2, the RSC parameter on $\mathcal{C}$ is unknown. These observations motivate the study of the averaged iterates of PSGD, and the next theorem confirms that these motivations hold true in our setting. To establish this result, we also impose a mild regularity condition on $f_\mu$, requiring it to be $\alpha$-Hölder continuous for some $\alpha \in (0, 1]$.

**Theorem 4.5** (Averaged iterates)**.** *Under Assumptions A or B, and assuming that $f_\mu$ is $\alpha$-Hölder with $\alpha \in (0, 1]$, for any decay schedule of the form $\gamma_n = \gamma_1/n^b$ with $\gamma_1 > 0$ and $b \in \left(\frac{1}{1+\alpha}, 1\right)$, noting $\lambda$ the second smallest eigenvalue of $H$ at the optimum, we have the convergence rate*

$$\mathbb{E}[\|\overline{\mathbf{g}}_n - \mathbf{g}^*\|_v^2] = \frac{1}{\lambda^2(n+1)} + o\left(1/n\right).$$

*Without assuming $f_\mu$ to be $\alpha$-Hölder, and for $b \in (1/2, 1)$, we still obtain*

$$\mathbb{E}[\|\overline{\mathbf{g}}_n - \mathbf{g}^*\|_v^2] = \mathcal{O}\left(1/n^b\right).$$

As we can see, the convergence rate depends on the constant $\lambda > 0$, which can be understood as the local strong convexity of $H$ at the optimum. Notably, there is existing literature on $\lambda^\varepsilon$ for the entropy-regularized semi-dual problem when $\mu$ has bounded support ([13], Theorem 3.2, [12], Proposition 5.1). These results can be extended to estimate $\lambda_*$ by letting the regularization parameter vanish.

### 4.2.1 Estimation of the OT cost

Building on the convergence rate of PSGD, we derive the corresponding rate for the OT cost estimation.

**Corollary 4.6.** *Under Assumption A or B, $H$ is $C^1$-smooth and uniformly bounded, so we have*
$$H(\mathbf{g}) - H(\mathbf{g}^*) = \mathcal{O}\left(\|\mathbf{g} - \mathbf{g}^*\|_v^2\right).$$
*Therefore, the OT cost exhibits the same convergence rate as the OT potential.*

As we can see, this result establishes a (nearly) $\mathcal{O}(1/n)$ convergence rate for the estimation of the OT cost, matching the rate derived in Theorems 4.4 and 4.5. In particular, for MTW costs, a faster convergence rate of $\mathcal{O}(1/n)$ is achievable, in contrast to the $\mathcal{O}(1/\sqrt{n})$ rate from Theorem 3.2 in the general setting.

## 5 OT cost and map estimation with PSGD

### 5.1 Minimax estimation of the OT map and Brenier potential

Having the convergence rate of PSGD to the Brenier potential $\mathbf{g}^*$, we study here the convergence of the map estimate $T(\mathbf{g}) : x \mapsto x - \nabla \mathbf{g}^c(x)$. Note that, as soon as there exists $j \in [\![1, M]\!]$ such that $x$ is in the interior of $\mathbb{L}_j(\mathbf{g}^*) \cap \mathbb{L}_j(\mathbf{g})$, we have
$$T_{\mu,\nu}(x) = x - \nabla(\mathbf{g})^c(x) = y_j.$$

Therefore, a result on the quantitative stability of the measure of Laguerre cells is sufficient to establish a convergence rate for the map estimator obtained from PSGD. Such a result was previously established in the compact case in [6]. Here, we extend their result to the quadratic Euclidean cost in the non-compact setting, leading to the following theorem.

**Theorem 5.1.** *Under Assumption A or B, the function $\mathbf{g} \mapsto \|T(\mathbf{g}) - T_{\mu,\nu}\|_{L^2(\mu)}^2$ is Lipschitz with respect to the infinity norm $\|\cdot\|_\infty$.*

As a corollary, we retrieve the convergence rate of our map estimator with PSGD.

**Corollary 5.2.** *Under the same assumptions as Theorem 4.4, taking $\hat{\mathbf{g}}_n \in \{\mathbf{g}_n, \overline{\mathbf{g}}_n\}$*
$$\mathbb{E}[\|T(\hat{\mathbf{g}}_n) - T_{\mu,\nu}\|_{L^p(\mu)}] = \mathcal{O}\left(1/n^{b/2}\right).$$
*If in addition, $f_\mu$ is $\alpha$-Hölder with $\alpha \in (0, 1]$, taking $b \in (\frac{1}{1+\alpha}, 1)$, we have*
$$\mathbb{E}[\|T(\overline{\mathbf{g}}_n) - T_{\mu,\nu}\|_{L^p(\mu)}] = \mathcal{O}\left(M^2/\lambda\sqrt{n}\right).$$

Note that our dependence on $M$ might be conservative. However, we prove that the rate $\mathcal{O}(1/\sqrt{n})$ achieved by $T(\overline{\mathbf{g}}_n)$ is minimax optimal.

**Theorem 5.3.** *Fixing $c(x, y) = \frac{1}{2}\|x - y\|^2$ and $\nu = \frac{1}{2}\delta_{\{0\}} + \frac{1}{2}\delta_{\{1\}}$ and noting $\mathcal{P}_{Lip}(\mathbb{R})$ the set of probability measures on $\mathbb{R}$ with Lipschitz densities, we have*
$$\inf_{T^{(n)}} \sup_{\mu \in \mathcal{P}_{Lip}(\mathbb{R})} \mathbb{E}_\mu\left[\left\|T^{(n)} - T_{\mu,\nu}\right\|_{L^p(\mu)}^p\right] \gtrsim 1/\sqrt{n},$$
*where the infimum is taken over all maps $T^{(n)}$ constructed with the $n$ i.i.d samples of $\mu$.*

We recover the same minimax lower bound as in the two-sample setting considered in [31], where the target measure $\nu$ is also subsampled. This shows that, even though we have full information about the target measure, the asymptotic rates remain the same. However, we are able to achieve this rate in the non-batched setting, without the need to calibrate a regularization parameter as in [31], or to know the number of samples in advance. Note also that a direct corollary of Theorem 5.3 and Theorem 5.1 is that the convergence rate $\mathcal{O}(1/n)$ for the estimation of the Brenier potential, achieved by the averaged iterates of PSGD, is also minimax optimal.

# 6    Numerical experiments

In this section, we numerically verify our convergence rate guarantees through various examples. All experiments demonstrating convergence rates were repeated 20 times, and the error plots represent the averaged errors. We set the learning rate to $\gamma_1 = \text{Diam}(\mathcal{C})$, as suggested by the analysis in Theorem 3.2. The step decay parameter $b$ was set to $3/4$, unless stated otherwise. We find that this learning rate leads to robust results without requiring further tuning. For each example, we generate $\mathbf{g}^*$ randomly, and approximate the associated Laguerre cell measures $\mu(\mathbb{L}_i^c(\mathbf{g}^*))$. We then fix $w_i = \mu(\mathbb{L}_i^c(\mathbf{g}^*))$, such that $\mathbf{g}$ is optimal by the first-order condition. The Laguerre cells are estimated with $10^9$ samples. All experiments were repeated 10 times, and the average performance was reported. We consider the following three settings to evaluate our method:

**Example 1: Non-quadratic cost.** The cost to move mass is set to $\|\cdot\|^{1.5}$. We set $\mu$ as the uniform measure $\mathcal{U}([0,1]^{10})$ and take $M = 50$ points $y_1, \ldots, y_M$ uniformly in $[0,1]^{50}$. The projection set is then $\mathcal{C} = [-10^{3/4}, 10^{3/4}]^{50}$.

**Example 2: Non-compact case.** Here, $\mu$ has full support on $\mathbb{R}^{10}$ with cost $c(x,y) = \frac{1}{2}\|x-y\|^2$. We choose $\mu$ with density $f_\mu(x) \propto (1+\|x\|)^{-d-3}$, satisfying (B1-3). As in Example 1, we sample $M = 50$ points in $[0,1]$. The projection set $\mathcal{C} = [-5,5]^{50}$ since $K = B(0,1)$ satisfies Lemma 3.1.

**Example 3: Non-smooth source measure.** We define $\mu$ with density $f_\mu(x) = 1/(2\sqrt{x})\mathbf{1}_{x \in (0,1]}$, which satisfies a $(1,1)$-Poincaré-Wirtinger inequality but is not $\alpha$-Hölder. We took $M = 10$ points uniformly in $[0,1]$. Since our results do not guarantee acceleration for non-Hölder densities, we set $b = 0.9$ for PSGD, as our analysis recommends $b$ close to 1 for the best rate. The projection set is $\mathcal{C} = [-1,1]^{10}$.

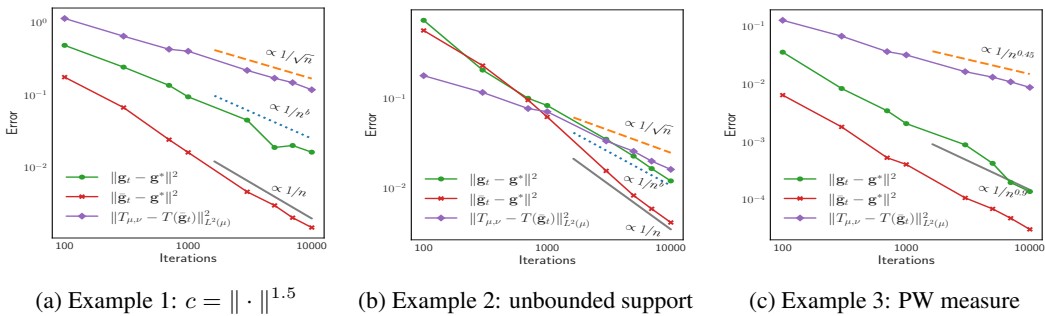

(a) Example 1: $c = \|\cdot\|^{1.5}$     (b) Example 2: unbounded support     (c) Example 3: PW measure

Figure 1: Convergence rates of our OT potential and OT map estimators across different settings.

As illustrated in Figure 1, our theoretical claims are well supported by empirical results. (i) Our convergence guarantees are matched: PSGD exhibits the expected convergence behavior across all three settings. In particular, we observe a rate of $\mathcal{O}(1/\sqrt{n})$ in Examples 1 and 2, and a rate of $\mathcal{O}(1/n^{0.45})$ in Example 3 for estimating the OT map. The exponent $0.45$ corresponds to $b/2$ with $b = 0.9$, which aligns with our theoretical guidance to select $b$ close to 1 when the source measure is not $\alpha$-Hölder regular but still satisfies a (PW) condition. (ii) Averaging yields optimal rates: In Examples 1 and 2, averaging leads to the optimal rate $\mathcal{O}(1/\sqrt{n})$ for the OT map without requiring $b = 1$. This confirms our theory, which remains robust to the choice of $\gamma_1 > 0$ and $b \in (1/2, 1)$, thanks to averaging, for achieving this minimax rate. (iii) We achieve minimax rates across our settings: Our results match the minimax rate $\mathcal{O}(1/\sqrt{n})$ and extend the findings of [31], who established similar behavior in the compact case with quadratic cost. Importantly, we observe that the estimation of the OT map avoids the curse of dimensionality in both compact (MTW cost) and non-compact (quadratic cost) semi-discrete settings.

# 7    Conclusion and Discussion

We studied SGD-based solvers for the semi-discrete optimal transport (OT) problem, focusing on settings where only one or a few samples are available per iteration. These solvers are widely used in machine learning applications involving semi-discrete OT, yet their theoretical understanding remains incomplete. Our work bridges this gap by proving that such methods can consistently estimate both

the OT cost and the OT map across a broad class of settings. Focusing on PSGD, we established minimax-optimal rates for estimating the OT map under MTW-type costs on compact domains, and under the quadratic Euclidean cost on both compact and non-compact domains. These results rely on novel convergence guarantees and structural properties of the semi-dual OT functional, stemming from the projection set we introduced and the enhanced properties of $H$ we obtained thanks to the restriction of our minimization space.

**Future directions: exploiting RSC with adaptive methods.** Our analysis suggests a promising avenue for future work: leveraging RSC to improve the performance of adaptive SGD methods such as using S-Adam [36] with projection for OT. While Adam is commonly used in semi-discrete OT (especially in generative modeling), it often suffers from convergence plateaus due to its fixed step size. In contrast, S-Adam incorporates a decaying learning rate and is specifically tailored for strongly convex objectives. Despite lacking theoretical guarantees under RSC, our empirical results (Figure 2) show that Projected S-Adam better exploits the local geometry of the semidual problem, outperforming Projected Adam in practice. Formalizing these observations and extending our theory to include adaptive methods, while challenging, remains a compelling direction for future research.

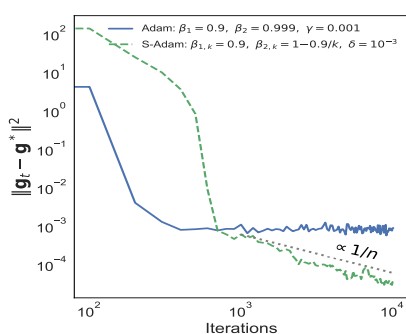

Figure 2: S-Adam outperforms Adam on Ex. 1, avoiding convergence plateau.

# 8 Acknowledgements

The work of François-Xavier Vialard is partly supported by the Bézout Labex (New Monge Problems), funded by ANR, reference ANR-10-LABX-58.

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

# Appendix

## Table of Contents

# Notations

- $\mathbb{R}^*$: The set $\mathbb{R} \setminus \{0\}$ (i.e., real numbers excluding zero).

- $\|\cdot\|$: The Euclidean norm. item $\lambda_{\mathbb{R}^d}$: The Lebesgue measure on $\mathbb{R}^d$.

- **Diameter**: For $\mathcal{C} \subset \mathbb{R}^d$, we define its diameter as:

$$D_{\mathcal{C}} := \sup\{\|x - y\| \mid x, y \in \mathcal{C}\}.$$

- **Hausdorff measure**: in $\mathbb{R}^d$, for $c \leq d$, $\mathcal{H}^c$ refers to the $c$-dimensional Hausdorff measure.

- **Indicator function**: For a set $A \subset \mathbb{R}^d$, $\mathbb{1}_A(x)$ is defined as $\mathbb{1}_A(x) = 1$ if $x \in A$, and $\mathbb{1}_A(x) = 0$ otherwise.

- **Component-wise Minimum**: For $v \in \mathbb{R}^d$: $v_{\min} := \min_{1 \leq j \leq d} v_j$.

- **Special Vectors**:

    - $\mathbf{1}_M = (1, \ldots, 1) \in \mathbb{R}^M$.
    - $\mathbf{0}_M = (0, \ldots, 0) \in \mathbb{R}^M$.
    - $\mathbf{e}_j \in \mathbb{R}^M$, for any $1 \leq j \leq M$, is the vector with zeros except for the $j$-th entry, which is equal to 1.

- **Probability Measures**:

    - $\mathcal{P}(\mathbb{R}^d)$: The set of probability measures on $\mathbb{R}^d$.
    - For $\rho \in \mathcal{P}(\mathbb{R}^d)$, $\mathrm{Supp}(\rho)$ denotes its support.

- **Asymptotic Orders**:

    - $\mathcal{O}(\cdot)$ and $o(\cdot)$: Standard approximation orders.
    - $f \lesssim g$ means there exists a constant $C > 0$ such that $f(\cdot) \leq Cg(\cdot)$.
    - $a \asymp b$ means both $a \lesssim b$ and $b \lesssim a$.

- **Filtration**: We denote by $\mathcal{F}_n$ the filtration generated by the sample $X_1, \ldots, X_n \overset{\text{iid}}{\sim} \mu$, i.e.,

$$\mathcal{F}_n = \sigma(X_1, \ldots, X_n), \quad n \geq 1.$$

- **Sets**:

    - For any $\varepsilon > 0$, define:

$$K_\varepsilon := \left\{ \mathbf{g} \in \mathbb{R}^M \mid \forall i \in [\![1, M]\!], \mu\left(\mathbb{L}_i(\mathbf{g})\right) \geq \varepsilon \right\}.$$

    - Define:

$$K_+ := \left\{ \mathbf{g} \in \mathbb{R}^M \mid \forall i \in [\![1, M]\!], \mu\left(\mathbb{L}_i(\mathbf{g})\right) > 0 \right\}.$$

- **Density of an Absolutely Continuous Measure**: For an absolutely continuous measure $\rho$ on $\mathbb{R}^d$, we denote its density w.r.t. the Lebesgue measure by $f_\rho$.

- **Orthogonal of Vect$(\mathbf{1})$**:

    - For any $\mathbf{g}, \mathbf{g}' \in \mathbb{R}^M$, we define:

$$\|\mathbf{g} - \mathbf{g}'\|_v = \|\mathrm{Proj}_{\mathbf{1}^\perp}(\mathbf{g} - \mathbf{g}')\|^2.$$

    - Inner product in this space:

$$\langle \mathbf{g}, \mathbf{g}' \rangle_v = \langle \mathrm{Proj}_{\mathbf{1}^\perp}(\mathbf{g}), \mathrm{Proj}_{\mathbf{1}^\perp}(\mathbf{g}') \rangle.$$

- **Strong Convexity**: We discuss the strong convexity of the semi-dual function $H$ when the strong convexity holds on the orthogonal complement of $\mathbf{1}$.

# A  Further details on our Assumptions

## A.1  The Ma-Trundinger-Wang Properties

In the semi-discrete setting, the class of cost functions verifying the Ma–Trudinger–Wang (MTW) properties [25] is defined as the set of cost functions satisfying the following conditions: (Reg), (Twist), and Loeper's condition (QC) detailed below.

$$c\left(\cdot, y_i\right) \in C^2(\mathrm{Supp}(\mu)), \forall i \in \{1, \dots, M\} \tag{Reg}$$

$$\nabla_x c\left(x, y_i\right) \neq \nabla_x c\left(x, y_k\right), \forall x \in \mathrm{Supp}(\mu), i \neq k \tag{Twist}$$

**Definition A.1** (Loeper's condition)**.** We say $c$ satisfies *Loeper's condition* if, for each $i \in \{1, \dots, M\}$, there exists a convex set $X_i \subset \mathbb{R}^d$ and a $C^2$ diffeomorphism $\exp_i^c(\cdot) : X_i \to \mathrm{Supp}(\mu)$ such that

$$\forall t \in \mathbb{R},\ 1 \leq k, i \leq M,\ \{p \in X_i \mid -c(\exp_i^c(p), y_k) + c(\exp_i^c(p), y_i) \leq t\} \text{ is convex}. \quad \text{(QC)}$$

**Definition A.2** (*c*-convexity)**.** We say that $X \subset \mathbb{R}^d$ is $c$-convex if $(\exp_i^c)^{-1}(X)$ is a convex set for every $i \in \{1, \dots, N\}$.

For a detailed discussion on this class of cost functions and their implications in the semi-discrete optimal transport framework, we refer the reader to Section 1.5 of [21]

## A.2  The Poincaré-Wirtinger Inequality

A probability measure $\rho = w(x)\,dx$ on a domain $\Omega \subset \mathbb{R}^d$ is said to satisfy the *weighted Poincaré–Wirtinger inequality* (PW) if

$$\int_\Omega |f - \mathbb{E}_\rho[f]|\,d\rho \ \leq\ C_{\mathrm{PW}} \int_\Omega |\nabla f|\,d\rho, \qquad \forall f \in C^1(\Omega).$$

The existence of a finite constant $C_{\mathrm{PW}}$ provides a quantitative connectedness of the source measure and is necessary for $H$ to be locally strongly convex. We provide here two examples of measures satisfying (PW). Notably, Example 1 shows that non-degenerate Gaussians, mixture of non-degenerate Gaussians and Student distributions satisfy (PW), when we take their restrictions on any ball $B(0, R)$, $R > 0$.

**Example 2.** (bounded support, density bounded above and below) Let $\Omega$ be bounded, connected, $\alpha$-Holder with $\alpha \in (0, 1]$, and assume $0 < m \leq w(x) \leq M < \infty$ almost everywhere.

Note that the assumption of the support being bounded from above and below is classical in the semi-discrete OT literature, as in [13, 31].

**Example 3.** (Annular support with radial concave profile, ([21], Proposition A.1)) Let $0 < r < R$, and let $\overline{\rho} \in \mathcal{C}^0([0, R])$ be a nonnegative function such that $\overline{\rho}(s) = 0$ for $s \in [0, r]$, and $\overline{\rho}$ is concave on $[r, R]$, with

$$\int_r^R \overline{\rho}(s)\,ds = 1.$$

Define the probability measure $\rho$ on the annulus $X := B(0, R) \subset \mathbb{R}^d$ by

$$\rho(x) = \frac{1}{\|x\|^{d-1}\omega_{d-1}}\,\overline{\rho}(\|x\|),$$

where $\omega_{d-1}$ denotes the surface volume of the unit sphere $\mathbb{S}^{d-1}$. Then $\rho$ satisfies the weighted Poincaré–Wirtinger inequality for some positive constant.

# B  Properties of the semi-discrete OT problem

**Regularity properties of $H$**

In the main article, we concisely presented the regularity properties of the function $H$. In this section, we provide a more detailed breakdown of these properties, organizing them into sub-properties and referring to the corresponding proofs in the quadratic case with unbounded support (assumptions B1–B3).

- **Differentiability:** The function $H$ is differentiable on the entire space $\mathbb{R}^M$, and we denote its gradient by $\nabla H$ (see Proposition B.5).

- **Local $C^2$ regularity:** There exists a radius $r > 0$ such that $H$ is twice continuously differentiable ($C^2$) on the ball $B(\mathbf{g}^*, r)$ (see Proposition B.5).

- **Local strong convexity:** The function $H$ is strongly convex on the ball $B(\mathbf{g}^*, r)$ (see Proposition B.12).

- **Hölder continuity of the Hessian:** If the density $f_\mu$ is $\alpha$-Hölder continuous for some $\alpha \in (0, 1]$, then the Hessian of $H$ inherits this regularity and is also $\alpha$-Hölder continuous (see Corollary B.11).

## B.1 Known results in the compact case

In this section, we recall known properties of the semi-dual semi-discrete problem when the support of the source measure $\mu$ is $c$-convex and contained within a compact set and $c$ is a cost satisfying the MTW properties. We will then extend these results to the non-compact case for the quadratic cost.

Here, we fix $\varepsilon > 0$ and recall that $K_\varepsilon := \left\{\mathbf{g} \in \mathbb{R}^M : \forall i \in [\![1, M]\!], \mu\left(\mathcal{L}_i(\mathbf{g})\right) \geq \varepsilon\right\}$. The two theorems presented below are taken from [21] and have been adapted to our notation. We emphasize that the authors of [21] considered the semi-dual OT problem as a concave problem, studying the objective function $-H$ instead of $H$ under their notation. For a better understanding of the constants in their theorems, we refer the reader to their article.

**Proposition B.1** (Theorem 1.1 in [21]). *Let $\mu$ be an absolutely continuous density with bounded support included in $\mathbb{R}^d$, then the functional $H$ is $C^1$ smooth, its gradient is given by*

$$\nabla H(\mathbf{g})_i = -\mu(\mathbb{L}_i^c(\mathbf{g})) + w_i \ ,$$

*and its Hessian by*

$$(i \neq j) \qquad \nabla^2 H(\mathbf{g})_{ij} = -\int_{\mathbb{L}_i(\mathbf{g}) \cap \mathbb{L}_j(\mathbf{g})} \frac{f_\mu(x)}{\|y_i - y_j\|} \, \mathrm{d}\mathcal{H}^{d-1}(x),$$

$$\nabla^2 H(\mathbf{g})_{ii} = -\sum_{j \neq i} \nabla^2 H(\mathbf{g})_{ij} \ .$$

**Proposition B.2** (Theorem 5.1 in [21]). *Under the assumption (A1), that $\mu$ satisfies a weighted (1,1)-Poincaré–Wirtinger inequality, there exists a constant $\lambda$ such that for any $\mathbf{g} \in K_\varepsilon$, the second smallest eigenvalue of $\nabla^2 H(\mathbf{g})$, denoted $\lambda_2(\nabla^2 H(\mathbf{g}))$, satisfies*

$$\lambda_2(\nabla^2 H(\mathbf{g})) > \lambda.$$

*That is, $H$ is strongly convex on $K_\varepsilon$, considering the problem on $\mathbf{1}^\perp$.*

**Theorem B.3** (Theorem 1.3 in [21]). *If $\mu$ has its density $f_\mu$ in $C^{0,\alpha}(Supp(\mu))$. Then, the functional $H$ is $C^{2,\alpha}$ on the set*

$$K_\varepsilon := \left\{\mathbf{g} \in \mathbb{R}^M, \forall i, \mu\left(\mathbb{L}_i(\mathbf{g})\right) > \varepsilon\right\} \ ,$$

Lastly, we state a result concerning the quantitative stability of Laguerre cells, as presented in [6].

**Lemma B.4** (Lemma 5.5 in [6]). *Under the same assumptions as in Proposition B.2, for $\mathbf{g}, \mathbf{g}' \in \mathbb{R}^M$, we have*

$$\mu(\mathbb{L}_i^c(\mathbf{g}) \setminus \mathbb{L}_i^c(\mathbf{g}')) \lesssim M\|\mathbf{g} - \mathbf{g}'\|_\infty, \qquad \forall i \in [\![1, M]\!] \ .$$

Once again, we refer to [6] for a more detailed understanding of the constant involved in this lemma.

## B.2 New properties for the non-compact case with the quadratic Euclidean cost

In this section, we give second order properties of the semi-dual when the source measure is not supported on a compact. In the compact case, this properties are already known as discussed in Appendix B.1.

For the reader's convenience, we recall the Assumptions that we made for the non-compact case.

**Assumption** (Non-compact case) **(B1)** The cost is quadratic: $c(x, y) = \frac{1}{2}\|x - y\|^2$ and the measure $\mu$ has a finite second-order moment.

**(B2)** There exists a compact set $K \subset \mathbb{R}^d$ with $\mu(K) \geq 1 - \frac{1}{4}w_{\min}$, such that the probability measure $\mu_K$ with density $f_{\mu_K}(x) := c_K f_\mu(x)\mathbf{1}_K(x)$ satisfies a Poincaré-Wirtinger inequality.

**(B3)** The density $f_\mu$ satisfies the following integrability and regularity condition: for $R > 1$ and $r \geq 1$, define
$$f_\mu^R := f_\mu \cdot \mathbf{1}_{\|x\| \leq R}, \quad f_\mu^{R+r} := f_\mu \cdot \mathbf{1}_{R+(r-2) \leq \|x\| \leq R+r}.$$
Assume there exist $C > 0$ and a modulus of continuity $\omega$ such that for all $\delta > 0$,

$$\sum_{r=0}^\infty (R + r)^{d-1} \omega_{f_\mu}^{R+r}(\delta) \leq C\,\omega(\delta), \quad \sum_{r=0}^\infty (R + r)^{d-1} C_{f_\mu}^{R+r} < \infty, \tag{6}$$

where $C_{f_\mu}^{R+r} := \sup_{x \in \mathbb{R}^d} f_\mu^{R+r}(x)$ and $\omega_{f_\mu}^{R+r}$ is the modulus of continuity of $f_\mu^{R+r}$.

**Additional notation.** Since here the cost is fixed, we define the Laguerre cells by $\mathbb{L}_i(\mathbf{g})$ instead of $\mathbb{L}_i^c(\mathbf{g})$.

### B.2.1 Definition and regularity of the Hessian

**Proposition B.5.** *Under Assumptions (B1) and (B3), the semi-dual $H$ is differentiable everywhere, and its gradient is given by*
$$\nabla H(\mathbf{g})_i = \mu(\mathbb{L}_i(\mathbf{g})) - w_i\,.$$

*Moreover $H$ is $C^2$ smooth on $K_{w_{\min}/2} \cap \mathcal{C}$ and its Hessian is given by*

$$(i \neq j) \qquad \nabla^2 H(\mathbf{g})_{ij} = -\int_{\mathbb{L}_i(\mathbf{g}) \cap \mathbb{L}_j(\mathbf{g})} \frac{f_\mu(x)}{\|y_i - y_j\|}\,\mathrm{d}\mathcal{H}^{d-1}(x),$$
$$\nabla^2 H(\mathbf{g})_{ii} = -\sum_{j \neq i} \nabla^2 H(\mathbf{g})_{ij}\,.$$

*Proof.* **Definition of the gradient.** The proof follows the lines of the proof of Theorem 4.1 of [21] and extend this result to the non-compact case. If $x$ is in the interior of Laguerre cell $\mathbb{L}_j(\mathbf{g})$, we have
$$\nabla_{\mathbf{g}} h(\mathbf{g}, x) = \mathbb{1}_{i=j} - w_i.$$
where we recall that $h(\mathbf{g}, x) = -\mathbf{g}^c(x) - \sum_{i=1}^M w_j g_j$. Since the boundaries of the Laguerre cells are defined by the intersections of $M$ hyperplanes, they form a negligible set with respect to the measure $\mu$. As a result, the gradient definition of $H$ follows immediately.

**Definition of the Hessian.** Fix $i \in [\![1, M]\!]$. We aim to prove the differentiability of the measure $\mu(\mathbb{L}_i(\mathbf{g}))$ with respect to $g_j$ and that its differential is defined by:

$$\frac{\partial \mu(\mathbb{L}_i(\mathbf{g}))}{\partial g_j} = -\int_{\mathbb{L}_i(\mathbf{g}) \cap \mathbb{L}_j(\mathbf{g})} \frac{f_\mu(x)}{\|y_i - y_j\|}\,\mathrm{d}\mathcal{H}^{d-1}(x), \quad j \neq i,$$
$$\frac{\partial \mu(\mathbb{L}_j(\mathbf{g}))}{\partial g_j} = -\sum_{i \neq j} \frac{\partial \mu(\mathbb{L}_i(\mathbf{g}))}{\partial g_j}.$$

This gives us exactly the line $\left(\nabla^2 H(\mathbf{g})_{1i}, \ldots, \nabla^2 H(\mathbf{g})_{Mi}\right)$ of the Hessian.

**Suppose** $i \neq j$**.**

Suppose $\delta \geq 0$. Defining $h_{ij}(x) := \frac{1}{2}\|x - y_i\|^2 - \frac{1}{2}\|x - y_j\|^2 - g_i + g_j$, note that we have $\mathbb{L}_i(\mathbf{g}) = \cap_j h_{ij}^{-1}(]-\infty, 0])$.

We also have $\mathbb{L}_i(\mathbf{g} + \delta\mathbf{e}_j) = \mathbb{L}_i(\mathbf{g}) \setminus \left(\cap_{k \neq j} h_{ik}^{-1}(]-\infty, 0]) \cap h_{ij}^{-1}([-\delta, 0])\right)$. Note that $h$ is Lipschitz and for all $x$, $\|\nabla h(x)\| = \|y_j - y_i\|$. Moreover, under assumption (B3), we can use Lemma F.2,

which states that for all hyperplane $H$, $\int_H f_\mu d\mathcal{H}^{d-1} \lesssim 1$. Therefore, we can apply the coarea formula to pass from the second to the third equality below,

$$\mu\left(\mathbb{L}_i(\mathbf{g} + \delta\mathbf{e}_j)\right) = \mu(\mathbb{L}_i(\mathbf{g})) - \mu\left(\cap_{k\neq j} h_{ik}^{-1}(]-\infty, 0]) \cap h_{ij}^{-1}\left([-\delta, 0]\right)\right)$$

$$= \mu(\mathbb{L}_i(\mathbf{g})) - \int_{\cap_{k\neq j} h_{ik}^{-1}(]-\infty,0])\cap h_{ij}^{-1}([-\delta,0])} f_\mu(x)\mathrm{d}x$$

$$= \mu(\mathbb{L}_i(\mathbf{g})) - \int_{-\delta}^0 \int_{\cap_{k\neq j} h_{ik}^{-1}(]-\infty,0])\cap h_{ij}^{-1}(\{t\})} \frac{f_\mu(x)}{\|y_j - y_i\|}\mathrm{d}\mathcal{H}^{d-1}(x)\mathrm{d}t\ .$$

By analogy, for $\delta \leq 0$ we have:

$$\mathbb{L}_i(\mathbf{g} + \delta\mathbf{e}_j) = \mathbb{L}_i(\mathbf{g}) \cup \left(\cap_{k\neq j} h_{ik}^{-1}(]-\infty, 0]) \cap h_{ij}^{-1}\left([-\delta, 0]\right)\right)\ ,$$

Using the coarea formula gives:

$$\mu\left(\mathbb{L}_i(\mathbf{g} + \delta\mathbf{e}_j)\right) = \mu(\mathbb{L}_i(\mathbf{g})) + \int_0^{-\delta} \int_{\cap_{k\neq j} h_{ik}^{-1}(]-\infty,0])\cap h_{ij}^{-1}(\{t\})} \frac{f_\mu(x)}{\|y_j - y_i\|}\mathrm{d}\mathcal{H}^{d-1}(x)\mathrm{d}t\ .$$

Applying Lemma B.6, which is stated and proved later in the appendix, the integrand defined above is continuous on $K_{w_{\min}/2}\cap\mathcal{C}$. As a consequence, we can apply the Fundamental Theorem of Calculus and justify the limit in

$$\lim_{\delta\to 0^-} \frac{\mu(\mathbb{L}_i(g + \delta\mathbf{e}_j)) - \mu(\mathbb{L}_i(\mathbf{g}))}{\delta} = -\int_{\cap_{k\neq j} h_{ik}^{-1}(]-\infty,0])\cap h_{ij}^{-1}(\{0\})} f_\mu(x)\frac{1}{\|y_j - y_i\|}\,\mathrm{d}\mathcal{H}^{d-1}(x)\ .$$

By symmetry

$$\lim_{\delta\to 0^+} \frac{\mu(\mathbb{L}_i(g + \delta\mathbf{e}_j)) - \mu(\mathbb{L}_i(\mathbf{g}))}{\delta} = -\int_{\cap_{k\neq j} h_{ik}^{-1}(]-\infty,0])\cap h_{ij}^{-1}(\{0\})} f_\mu(x)\frac{1}{\|y_j - y_i\|}\,\mathrm{d}\mathcal{H}^{d-1}(x)\ .$$

Since $\cap_{k\neq j} h_{ik}^{-1}(]-\infty, 0]) \cap h_{ij}^{-1}(\{0\}) = \mathbb{L}_i(\mathbf{g}) \cap \mathbb{L}_j(\mathbf{g})$, we thus obtain

$$F_{ij}(\mathbf{g}) := \frac{\partial\mu(\mathbb{L}_i(\mathbf{g}))}{\partial g_j} = -\int_{\mathbb{L}_i(\mathbf{g})\cap\mathbb{L}_j(\mathbf{g})} \frac{f_\mu(x)}{\|y_i - y_j\|}\,d\mathcal{H}^{d-1}(x),\quad j\neq i\ .$$

**Suppose** $i = j$. No matter $\mathbf{g} \in \mathbb{R}^M$, the Laguerre cells verifies $\mu\left(\cup_i \mathbb{L}_j(\mathbf{g})\right) = \mu(\mathbb{R}^d) = 1$. Therefore,

$$\frac{\partial}{\partial g_j}\sum_{i=1}^M \mu\left(\mathbb{L}_i(\mathbf{g})\right) = \frac{\partial}{\partial g_j}1 = 0.$$

This equality gives $\frac{\partial}{\partial g_j}\mu\left(\mathbb{L}_j(\mathbf{g})\right) = -\sum_{i\neq j}\frac{\partial}{\partial g_j}\mu\left(\mathbb{L}_i(\mathbf{g})\right)$.

$\square$

**Lemma B.6.** *Under assumption (B1) and (B3), for any $\mathbf{g} \in K_{w_{\min}/2}\cap\mathcal{C}$ the function $F_{f_\mu}$ defined for all $t \in \mathbb{R}$ as*

$$F_{f_\mu}(t) = \int_{\cap_{k\neq j} h_{ik}^{-1}(]-\infty,0])\cap h_{ij}^{-1}(\{t\})} \frac{f_\mu(x)}{\|y_j - y_i\|}\,\mathrm{d}\mathcal{H}^{d-1}(x)\ .$$

*admits $\omega$ as modulus of continuity in some neighborhood of $\{0\}$.*

*Proof of Lemma B.6.* We keep the same notation as in [21]. Restricting ourselves to the quadratic cost, we get

$$\varepsilon_{nd} = \varepsilon_{tw} = \min_{i\neq j}\|y_i - y_j\|\ ,$$

$$C_{\nabla}^R = O(R)\ ,$$

$$C_{exp} = O(1)\ ,$$

$$C_{cond} = O(1)\ ,$$

$$C_{det} = O(1)\ .$$

In order to apply the results in [21], we need to extend Proposition 4.5 to measures with unbounded support in the case of the quadratic cost. We prove the following result:

**Proposition B.7.** *Consider $\mathcal{K}_\varepsilon := \{\mathbf{g} \in \mathbb{R}^{N-1} \mid \mu(\mathbb{L}_i(\mathbf{g})) > \varepsilon, \forall i = 1, \ldots, M\}$ for some $\varepsilon > 0$ sufficiently small and let $\mathcal{C}$ be the compact set of the projection. Then, there exists a positive constant $\delta_1$, such that for all $\mathbf{g} \in \mathcal{K}_\varepsilon \cap \mathcal{C}$ and all $p \in \mathbb{R}^d$ such that there exist $i \neq j$, for which $c(p, y_i) - c(p, y_0) = g_i - g_0$ and $c(p, y_j) - c(p, y_0) = g_j - g_0$, then*

$$\left( \frac{\langle y_i - y_0, y_j - y_0 \rangle}{\|y_i - y_0\| \|y_j - y_0\|} \right)^2 \leq 1 - \delta_1^2 . \tag{7}$$

*Remark* B.8. Note that the transversality *equation* (7) is independent of $p \in \mathbb{R}^d$. However, $p$ is still involved in the transversality *condition* by the value of the difference of the costs.

*Proof.* Fix an index $i$. The set of points $p \in \mathbb{R}^d$ such that $c(p, y_i) - c(p, y_0) = g_i - g_0$ defines a hyperplane in $\mathbb{R}^d$. The intersection of two such hyperplanes, denoted $H_{ij}(g)$, is, unless the hyperplanes are parallel, a codimension-2 affine subspace of $\mathbb{R}^d$.

For each such $H_{ij}(g)$, let $d_{ij}(g) \in \mathbb{R}^d$ be the orthogonal projection of the origin $0_d$ onto $H_{ij}(g)$. Since the vector of potentials $(g_i)_{i=0,\ldots,M-1}$ is bounded due to the projection set $\mathcal{C}$, the set

$$\{d_{ij}(g) \, : \, i \neq j \text{ admissible, and } g \in \mathcal{K}_\varepsilon \cap \mathcal{C}\}$$

is contained in a ball $B(0, R-1)$ for some sufficiently large $R > 0$.

Hence, for every $g \in \mathcal{K}_\varepsilon \cap \mathcal{C}$ and every admissible pair $(i, j)$, there exists a point $p \in H_{ij}(g)$ lying in the interior of the ball $B(0, R)$.

We now apply the transversality result from [21] to the measure

$$\mu_R := \mathbf{1}_{B(0,R)} \cdot \frac{\mu}{\mu(B(0, R))},$$

which is the normalized restriction of $\mu$ to $B(0, R)$. This result guarantees transversality of the hyperplanes $H_{ij}$ associated with potentials in the set $\mathcal{K}'_{\varepsilon'}$ (defined analogously for $\mu_R$ and some $\varepsilon' > 0$) over the whole space $\mathbb{R}^d$.

Choose $R$ such that $\mu(B(0, R)) \geq 1 - \varepsilon/2$. Then it is sufficient to take

$$\varepsilon' = \frac{\varepsilon/2}{1 - \varepsilon/2},$$

since one can check that $\mathcal{K}_\varepsilon \subset \mathcal{K}'_{\varepsilon'}$. This completes the proof. □

We aim to use the transversality result of Kitagawa et al. on a decomposition of $\mathbb{R}^d$ into balls centered at 0. To this goal, we now prove a transversality result at the boundary of $B(0, R)$ uniform for $R$ sufficiently large.

**Proposition B.9.** *Consider $\mathcal{K}_\varepsilon := \{\mathbf{g} \in \mathbb{R}^{N-1} \mid \mu(\mathbb{L}_i(\mathbf{g})) > \varepsilon, \forall i = 1, \ldots, M\}$ for some $\varepsilon > 0$ sufficiently small and let $\mathcal{C}$ be the compact set of the projection. Then, for any positive constant $\delta_2$ there exists $R$ sufficiently large such that for all $\mathbf{g} \in \mathcal{K}_\varepsilon \cap \mathcal{C}$ and all $p \in \mathbb{R}^d$: if there exists $i$ such that $c(p, y_i) - c(p, y_0) = g_i - g_0$ for some $p \in \partial B(0, R)$, then*

$$\left\langle \frac{p}{\|p\|}, \frac{y_i - y_0}{\|y_i - y_0\|} \right\rangle \leq 1 - \delta_2^2 . \tag{8}$$

*Proof.* Rewrite the condition $c(p, y_i) - c(p, y_0) = g_i - g_0$ as $\langle p, y_i - y_0 \rangle = g_i - g_0$, dividing by $\|p\|$ we get

$$\left\langle \frac{p}{\|p\|}, y_i - y_0 \right\rangle = \frac{g_i - g_0}{\|p\|} . \tag{9}$$

The right hand side tends to 0 uniformly with $R$ since $g_i - g_0$ lies in a compact set. The conclusion follows directly. □

The rest of the proof follows the lines of the proofs Appendix B of [21] with $\varepsilon_{tr} = \delta_1$ as in Proposition B.7. From Proposition B.9 applied to $\delta_2 = \delta_1$ there exists $R > 1$ such as the transversality condition on the boundary (8) holds for every $B(0, R + r)$ and the same $\delta_1$ for every $r \geq 0$. Let us fix such $R > 1$ so that Assumption (5) is also satisfied.

The next step is to decompose the integral using a partition of unity. We define the sequence of functions $(\varphi_r)_{r \geq 0}$ by

$$\varphi_r(x) = \begin{cases} (R - \|x\|)_+ \wedge 1, & \text{if } r = 0, \\ (\|x\| - (R + r - 2))_+ \wedge (R + r - \|x\|)_+, & \text{if } r \geq 1, \end{cases} \qquad x \in \mathbb{R}^d.$$

An illustration of the functions $\varphi_r(x)$ defined above is shown below:

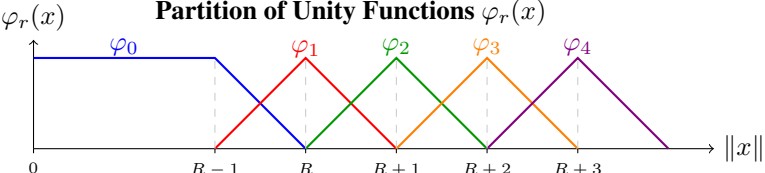

By definition, $\sum_{r \geq 0} \varphi(x) = 1$, for all $x \in \mathbb{R}^d$, and every $\varphi_r$ is supported on $\{R + r - 2 \leq \|x\| \leq R + r\}$ for every $r \geq 1$, $\varphi_0$ being supported by $B(0, R)$. Moreover $\varphi_r$ is Lipschitz continuous and we denote $\omega_{\varphi_r}$ its modulus of continuity on its support. The moduli of continuity satisfy $\omega_{\varphi_r}(\delta) \leq |\delta|$, $\delta > 0$ $r \geq 0$. Then, writing $S_{i,j,t} = \cap_{k \neq j} h_{ik}^{-1}(]-\infty, 0]) \cap h_{ij}^{-1}(\{t\})$ for conciseness, we decompose the integral

$$\int_{S_{i,j,t}} \frac{f_\mu(x)}{\|y_j - y_i\|} d\mathcal{H}^{d-1}(x) = \sum_{r=0}^{\infty} \int_{S_{i,j,t}} \frac{f_\mu(x)\varphi(x)}{\|y_j - y_i\|} d\mathcal{H}^{d-1}(x). \qquad (10)$$

We apply Proposition B.1 of [21] to each term of the decomposition. We recall below his crucial result, tracking the order of the constants established by [21].

**Proposition B.10.** *Let $\sigma$ be a continuous non-negative function on $B(0, R)$ bounded by $\sigma_\infty$ and with modulus of continuity $\omega_\sigma$. Let the functions $h_{ij}$ satisfy the transversality conditions (7) and (8) with the same constant $\varepsilon_{tr} > 0$. Then*

$$F_\sigma(t) := \int_{\cap_{k \neq j} h_{ik}^{-1}(]-\infty, 0]) \cap h_{ij}^{-1}(\{t\})} \frac{\sigma(x)}{\|y_j - y_i\|} d\mathcal{H}^{d-1}(x)$$

*has modulus of continuity*
$$\omega_{h_\sigma}(\delta) = C_1 \omega_\sigma(C_2 \delta) + C_3 |\delta|$$
*where $C_1 = O(\mathcal{H}^{d-1}(\partial B(0, R)))$, $\mathcal{H}^{d-1}(\partial B(0, R)) = O(R^{d-1})$, $C_2 = O(\varepsilon_{tw}^{-1}) = O(1)$ and $C_3 = O(\sigma_\infty C(d, 2R)\varepsilon_{tr}^{-4} + \mathcal{H}^{d-1}(\partial B(0, R)))$, where $C(d, 2R)$ defined in (3.5) of [21] satisfies $C(d, 2R) = O(R^{d-1})$.*

When applying Proposition B.10 to the continuous function $\sigma^{R+r}(x) := f_\mu(x)\varphi(x)$ we easily estimate $\sigma_\infty^{R+r} \leq C_{f_\mu}^{R+r}$ and $w_{\sigma^{R+r}} \leq \omega_{f_\mu}^{R+r} + C_{f_\mu}^{R+r}\omega_{\varphi_r}$. Using that $\omega_{\varphi_r}(\delta) \leq |\delta|$ we obtain:

$$\omega_{h_{\sigma^{R+r}}}(\delta) = O\left((R + r)^{d-1}\right) \left(\omega_{f_\mu}^{R+r}(O(1)\delta) + O(C_{f_\mu}^{R+r})\delta\right),$$

uniformly for every $\mathbf{g} \in \mathcal{K}_\varepsilon \cap \mathcal{C}$ on a neighborhood of $\{0\}$. Noticing that this neighborhood depends solely on the transversality properties that are common to every $r \geq 0$ and since

$$\omega_{F_{f_\mu}} = \sum_{r \geq 0} \omega_{h_{\sigma^{R+r}}}$$

from the decomposition in (10) the desired result follows under the assumption in (5). $\qquad \square$

**Corollary B.11.** *Under Assumptions (B1) and (B3), if moreover $f_\mu$ is $\alpha$-Hölder with $\alpha \in (0, 1]$, then the Hessian of the semi-dual $H$ is also $\alpha$-Hölder on $K_{w_{\min}/2} \cap \mathcal{C}$.*

*Proof.* Since $f_\mu$ is $\alpha$-Hölder, the function $f_\mu^{R+r}$ with $r \geq 0$ are also $\alpha$-Hölder and we note $\omega_{f_\mu}^{R+r} = \kappa_{f_\mu}^{R+r} \delta^\alpha$, applying Proposition B.10,

$$\omega_{h_{\sigma R+r}}(\delta) O\left((R+r)^{d-1}\right) \left(\omega_{f_\mu}^{R+r}(O(1)\delta) + O(C_{f_\mu}^{R+r})\delta\right)$$
$$= O((R+r)^{d-1})\kappa_{f_\mu}^{R+r} O(1)\delta^\alpha + ((R+r)^{d-1})O(C_{f_\mu}^{R+r})\delta .$$

By the summability conditions of assumption (B3), for some constants $C_1, C_2 > 0$, we have

$$\omega_{F_{f_\mu}} = \sum_{r \geq 0} \omega_{h_{\sigma R+r}}$$
$$= O(1)\delta^\alpha \sum_{r \geq 0} O((R+r)^{d-1})\kappa_{f_\mu}^{R+r} + \delta \sum_{r \geq 0} O(C_{f_\mu}^{R+r})((R+r)^{d-1})$$
$$\leq C_1 \delta^\alpha + C_2 \delta ,$$

Applying Proposition B.3 in [21] under our hypothesis shows that for $\mathbf{g}, \mathbf{g}' \in K_{w_{\min}/2} \cap \mathcal{C}$, there exists a constant $C$ depending on $\varepsilon_{tr}$ and $f_\mu$ such that we have

$$\left|\nabla^2 H(\mathbf{g}) - \nabla^2 H(\mathbf{g}')\right| \leq \omega_{F_{f_\mu}}(\|\mathbf{g} - \mathbf{g}'\|_\infty) + C\|\mathbf{g} - \mathbf{g}'\|$$

which gives the $\alpha$-Hölder regularity since $\omega_{F_{f_\mu}} \leq C_1 \delta^\alpha + C_2 \delta$. $\qquad\square$

### B.2.2 Local strong convexity with respect to the mass of Laguerre cells

**Proposition B.12.** *Under Assumptions (B1-B3) $H$ is strongly convex on $K_{\frac{1}{2}w_{\min}}$.*

*Proof.* Recall $K \subset \mathbb{R}^d$ the compact set such that $\mu(K) > 1 - \frac{1}{4}w_{\min}$ and $\mu_K$, the probability measure with density defined by $f_{\mu_K}(x) = c_K f_\mu(x)\mathbf{1}_K(x)$, with $c_K \in [1, 2]$, satisfies a weighted Poincaré-Wirtinger inequality.

We thus can use Proposition B.2, which states that the semi-dual between $\mu_R$ and $\nu$ is strongly-convex on $\text{Vect}_{\mathbf{1}}^\perp$ on the set

$$K_{\frac{1}{4}w_{\min}}^R := \left\{\mathbf{g} \in \mathbb{R}^M : \forall i \in [\![1, M]\!], \mu_R(\mathbb{L}_i^R(\mathbf{g})) \geq \frac{1}{4}w_{\min}\right\} .$$

This gives us that, for any $\mathbf{g} \in K_{\frac{1}{4}w_{\min}}^R$, there exists $\lambda_K > 0$, lower bounding the second smallest value of the semi-dual function between $\mu_R$ and $\nu$. This also gives us that for any $\mathbf{g} \in K_{\frac{1}{4}w_{\min}}^R$, the second smallest eigenvalue of the matrix $B$ defined by

$$(i \neq j) \qquad B_{ij} = -\int_{\mathbb{L}_i(\mathbf{g}) \cap \mathbb{L}_j(\mathbf{g})} \frac{f_\mu(x)\mathbf{1}_{B(0,R)}(x)}{\|y_i - y_j\|} \, d\mathcal{H}^{d-1}(x),$$
$$B_{ii} = -\sum_{j \neq i} B_{ij} ,$$

is lower bounded by $\lambda_K/c_R \geq \lambda_K/2 > 0$.

Since for any $\mathbf{g}$, and $i \neq j$, $\nabla^2 H(\mathbf{g})_{ij} \leq B_{ij}$ and that both $\nabla^2 H(\mathbf{g})$ and $B$ are Laplacian matrices, we apply Lemma F.1 to obtain that the second smallest eigenvalue of the hessian $\nabla^2 H$ is lower bounded by $\lambda_R/c_R$ on $K_{\frac{1}{4}w_{\min}}^R$.

For any $\mathbf{g} \in K_{w_{\min}/2}$, we thus have for all $i \in [\![1, M]\!], \mu(\mathbb{L}_i(\mathbf{g}) \cap B(0, R)) \geq \frac{1}{4}w_{\min}$. That is, $K_{\frac{1}{2}w_{\min}} \subset K_{\frac{1}{4}w_{\min}}^R$, which completes the proof. $\qquad\square$

### B.2.3 Quantitative stability of the Laguerre cells

**Lemma B.13.** *Under Assumptions (B1-B3), we have*

$$\mu(\mathbb{L}_i^c(\mathbf{g}) \setminus \mathbb{L}_i^c(\mathbf{g}')) \lesssim M\|\mathbf{g} - \mathbf{g}'\|_\infty, \qquad \forall i \in [\![1, M]\!] .$$

*Proof.* For all $j \in [\![1, M]\!]$, if $x$ is in the interior of $\mathbb{L}_j(\mathbf{g})$, we have

$$\nabla \mathbf{g}^c(x) = x - y_j. \tag{11}$$

Therefore, given $\mathbf{g}, \mathbf{g}' \in \mathbb{R}^M$, if there exists $j \in [\![1, M]\!]$ such that $x$ is the interior of $\mathbb{L}_j(\mathbf{g}) \cap \mathbb{L}_j(\mathbf{g}')$ we have

$$\nabla \mathbf{g}^c(x) = \nabla (\mathbf{g}')^c(x).$$

Moreover, since the support of $\nu$ is finite, we have

$$\sup_{x \in \mathbb{R}^d} \|\nabla \mathbf{g}^c(x) - \nabla(\mathbf{g}')^c(x)\| = \max_{i \neq j} \|y_i - y_j\| .$$

Hence, to bound the error of $T(\mathbf{g})(x) = x - \nabla \mathbf{g}^c(x)$ and $T_{\mu,\nu} = x - \nabla(\mathbf{g}^*)^c(x)$, we just need to bound the difference of measure of Laguerre cells made by $\mathbf{g}$ and $\mathbf{g}^*$. More generally, we now proceed here to bound the difference of measure of Laguerre cells between $\mathbf{g}, \mathbf{g}' \in \mathbb{R}^M$ fixed arbitrarily.

Our proof will follow some arguments from [21]. Let us fix $i \in [\![1, M]\!]$ and suppose $x \in \mathbb{L}_i(\mathbf{g}) \setminus \mathbb{L}_i(\mathbf{g}')$. The definition of the Laguerre cells implies that there is a $k \neq i$ such that $c(x, y_k) + \mathbf{g}'_k < c(x, y_i) + \mathbf{g}'_i$ while $c(x, y_i) + \mathbf{g}_i \leq c(x, y_k) + \mathbf{g}_k$. Combining these two inequalities yields to

$$\mathbf{g}'_k - \mathbf{g}'_i < c(x, y_i) - c(x, y_k) \leq \mathbf{g}_k - \mathbf{g}_i.$$

Hence, writing $f_{ik}(x) = c(x, y_i) - c(x, y_k)$, we have

$$\mathbb{L}_i(\mathbf{g}) \setminus \mathbb{L}_i(\mathbf{g}') \subset \bigcup_{k \neq i} f_{ik}^{-1}([\mathbf{g}'_k - \mathbf{g}'_i, \mathbf{g}_k - \mathbf{g}_i]) . \tag{12}$$

We now now bound $\mu\left(f_{ik}^{-1}([\mathbf{g}'_k - \mathbf{g}'_i, \mathbf{g}_i - \mathbf{g}_k])\right)$ using the coarea formula, using that for all $x$, $\|\nabla f_{ik}(x)\| = \|y_i - y_k\|$:

$$\mu\left(f_{ik}^{-1}([a,b])\right) = \int_{f_{ik}^{-1}([a,b])} \mathrm{d}\mu(x)$$
$$= \int_a^b \int_{f_{ik}^{-1}(\{t\})} \frac{1}{\|\nabla f_{ik}(x)\|} \mu(x) \mathrm{d}\mathcal{H}^{d-1}(x) \mathrm{d}t$$
$$= \int_a^b \int_{f_{ik}^{-1}(\{t\})} \frac{1}{\|y_i - y_k\|} \mu(x) \mathrm{d}\mathcal{H}^{d-1}(x) \mathrm{d}t .$$

Observe that for the quadratic cost, $f_{ik}(x) = \langle x, y_k - y_i \rangle + \|y_i\|^2 - \|y_k\|^2$ and so $f_{ik}^{-1}(\{t\}) = \left\{x \in \mathbb{R}^d, \langle x, y_k - y_i \rangle = t - \frac{1}{2}(\|y_i\|^2 + \|y_k\|^2)\right\}$ and is therefore a hyperplane. Applying Lemma F.2, there exists a constant $C$ such that, for any $i, k$ and $t$, we have

$$\int_{f_{ik}^{-1}(\{t\})} \frac{1}{\|y_i - y_k\|} \mu(x) \mathrm{d}\mathcal{H}^{d-1}(x) \leq C .$$

Therefore, we have

$$\mu\left(f_{ik}^{-1}([a,b])\right) \leq \int_a^b \frac{C}{\|y_i - y_k\|} \leq (b-a) \frac{C}{\|y_i - y_k\|} .$$

Since $\mathbf{g}_k - \mathbf{g}_i - (\mathbf{g}'_k - \mathbf{g}'_i) \leq 2\|\mathbf{g} - \mathbf{g}'\|_\infty$, by combining the above with (12) we conclude

$$\mu\left(\mathbb{L}_i(\mathbf{g}) \setminus \mathbb{L}_i(\mathbf{g}')\right) \leq \sum_{k \neq i} \mu\left(f_{ik}^{-1}([\mathbf{g}'_k - \mathbf{g}'_i, \mathbf{g}_k - \mathbf{g}_i])\right) \lesssim M\|\mathbf{g} - \mathbf{g}'\|_\infty. \tag{13}$$

$\square$

## B.3 New properties in both our compact and non compact settings

**Theorem 5.1.** Under Assumptions (A1) or (B1-3), the function $\mathbf{g} \mapsto \|T(\mathbf{g}) - T_{\mu,\nu}\|^2_{L^2(\mu)}$ is Lipschitz with respect to the infinity norm $\|\cdot\|_\infty$. Moreover, the Lipschitz constant grows at most quadratically in $M$.

*Proof.* Using that no matter $\mathbf{g} \in \mathbb{R}^M$, for any $x \in \mathbb{R}^d$

$$\|T_{\mu,\nu}(\mathbf{g}^*) - T_{\mu,\nu}(\mathbf{g})\| \leq \begin{cases} 0 & \text{if } x \in \mathbb{L}_i(\mathbf{g}^*) \cap \mathbb{L}_i(\mathbf{g}) \text{ for a certain } i \in [\![1,M]\!], \\ \max_{i \neq j} \|y_i - y_j\| & \text{else,} \end{cases}$$

and that for any $\mathbf{g}, \mathbf{g}' \in \mathbb{R}^M$, we have $\mu(\mathbb{L}_i(\mathbf{g}) \setminus \mathbb{L}_i(\mathbf{g}')) \lesssim M\|\mathbf{g} - \mathbf{g}'\|$ using Proposition B.4 in the compact case, or Proposition B.13, we have for any $p \in [1, \infty)$

$$\begin{aligned}
\|T_{\mu,\nu}(\mathbf{g}^*) - T_{\mu,\nu}(\mathbf{g})\|^p_{L_p(\mu)} &= \int_{\mathbb{R}^d} \|T_{\mu,\nu}(\mathbf{g}^*)(x) - T_{\mu,\nu}(\mathbf{g})(x)\|^p d\mu(x) \\
&= \sum_{i=1}^M \int_{\mathbb{L}_i(\mathbf{g}^*)} \|T_{\mu,\nu}(\mathbf{g}^*)(x) - T_{\mu,\nu}(\mathbf{g})(x)\|^p d\mu(x) \\
&\leq \sum_{i=1}^M \int_{\mathbb{L}_i(\mathbf{g}^*) \setminus \mathbb{L}_i(\mathbf{g})} \max_{i \neq j} \|y_i - y_j\|^p d\mu(x) \\
&\leq \sum_{i=1}^M \max_{i \neq j} \|y_i - y_j\|^p \mu(\mathbb{L}_i(\mathbf{g}^*) \setminus \mathbb{L}_i(\mathbf{g})) \\
&\lesssim M^2 \|\mathbf{g}^* - \mathbf{g}\|_\infty,
\end{aligned}$$

where we used $\mu(\mathbb{L}_i(\mathbf{g}) \setminus \mathbb{L}_i(\mathbf{g}^*)) \lesssim M\|\mathbf{g} - \mathbf{g}^*\|$ for the last line.

$\square$

**Lemma B.14.** *For any $\varepsilon > 0$, there exists $\mathbf{g}^* \in K_\varepsilon$ and $d_\varepsilon > 0$ such that $B(\mathbf{g}^*, d_\varepsilon) \subset K_\varepsilon$.*

*Proof.* This result is a simple application of the results on the quantitative stability of Laguerre cells stated in Proposition B.4 and Proposition B.13. $\square$

**Proposition B.15.** *(Hessian and Local strong convexity). Under Assumptions (A1) or (B1-3), $H$ is twice differentiable. Moreover, there exists a constant $\lambda > 0$ and a radius $r > 0$, such that for any $\mathbf{g} \in \mathbb{R}^M$, $\|\mathbf{g}^* - \mathbf{g}\|_v \leq r$ implies that the second smallest value of the Hessian $\nabla^2 H(\mathbf{g})$ is lower bounded by $\lambda$.*

*Proof.* The fact that $H$ is twice differentiable was already known in the compact case and is proven in Proposition B.5 for the non compact case under (B1-3).

Under assumptions (A1) and using Proposition B.2 or under assumptions (B1-3) and using Proposition B.12, $H$ is strongly convex on $K_{\frac{1}{2}w_{\min}}$. Applying Lemma B.14 concludes the proof. $\square$

**Proposition B.16.** *There exists $\eta > 0$, such that uniformly in $\mathbf{g} \in \mathcal{C}$, we have*

$$\langle \nabla H(\mathbf{g}), \mathbf{g} - \mathbf{g}^* \rangle_v \geq \eta \|\mathbf{g} - \mathbf{g}^*\|^2_v .$$

*Proof.* Observe that, by Proposition B.12, $H$ is locally strongly convex on the orthogonal of $\mathbf{1}$, on the set

$$K_{\frac{1}{2}w_{\min}} = \left\{ \mathbf{g} : \forall i \in [\![1,M]\!], \mu(\mathbb{L}_i(\mathbf{g})) \geq \frac{1}{2}w_{\min} \right\} .$$

Therefore, for any $\mathbf{g} \in K_{\frac{1}{2}w_{\min}}$, we have $\langle \nabla H(\mathbf{g}), \mathbf{g} - \mathbf{g}^* \rangle_v \geq \lambda \|\mathbf{g} - \mathbf{g}^*\|^2_v$.

Now, suppose $\mathbf{g} \in \mathcal{C} \setminus K_{\frac{1}{2}w_{\min}}$. Using Lemma B.14, we know that there exists $d_0 > 0$ such that $B(\mathbf{g}^*, d_0) \subset K_{\frac{1}{2}w_{\min}}$. Therefore, by the convexity of $H$, there exists $c > 0$ such that

$$\langle \nabla H(\mathbf{g}), \mathbf{g} - \mathbf{g}^* \rangle_v \geq H(\mathbf{g}) - H(\mathbf{g}^*) > c .$$

Defining

$$\lambda' := \inf_{\mathbf{g} \in \mathcal{C} \setminus K_{\frac{1}{2}w_{\min}}} \left\{ \frac{H(\mathbf{g}) - H(\mathbf{g}^*)}{\|\mathbf{g} - \mathbf{g}^*\|_v^2} \right\} > c/\mathrm{Diam}_2(\mathcal{C})^2,$$

we have for any $\mathbf{g} \in \mathcal{C} \setminus K_{\frac{1}{2}w_{\min}}$

$$\langle \nabla H(\mathbf{g}), \mathbf{g} - \mathbf{g}^* \rangle_v \geq \lambda' \|\mathbf{g} - \mathbf{g}^*\|_v^2 .$$

Taking $\eta = \min\{\lambda, \lambda'\}$ concludes the proof. □

## C  Proofs of the convergence rates of PSGD

### C.1  Fast convergence rates for MTW costs

#### C.1.1  Convergence of the non-averaged iterates.

**Theorem** 4.4 (Non averaged iterates). Under Assumptions (A1) or (B1-3) and for any decay step of the form $\gamma_n = \gamma_1/n^b$ with $\gamma_1 > 0, b \in (1/2, 1)$, we have the convergence rate

$$\mathbb{E}[\|\mathbf{g}_n - \mathbf{g}^*\|_v^2] = \mathcal{O}\left(\frac{\gamma_n}{\eta}\right) .$$

*Proof.* By definition of the gradient step at time $n + 1$, sampling $X_{n+1} \sim \mu$ and since $\mathbf{g}^* \in \mathcal{C}$, we have $\|\nabla_{\mathbf{g}} h(\mathbf{g}_n, X_{n+1})\|_v \leq 2$ a.s. and

$$\begin{aligned}
&\|\mathbf{g}_{n+1} - \mathbf{g}^*\|_v^2 \\
&= \|\mathrm{Proj}_{\mathcal{C}}(\mathbf{g}_n - \gamma_{n+1}\nabla_{\mathbf{g}} h(\mathbf{g}_n, X_{n+1})) - \mathbf{g}^*\|_v^2 \\
&\leq \|\mathbf{g}_n - \gamma_{n+1}\nabla_{\mathbf{g}} h(\mathbf{g}_n, X_{n+1}) - \mathbf{g}^*\|_v^2 \\
&\leq \left(\|\mathbf{g}_n - \mathbf{g}^*\|_v^2 - 2\gamma_{n+1}\langle \nabla_{\mathbf{g}} h(\mathbf{g}_n, X_{n+1}), \mathbf{g}_n - \mathbf{g}^* \rangle_v + \gamma_{n+1}^2 \|\nabla_{\mathbf{g}} h(\mathbf{g}_n, X_{n+1})\|_v^2\right) \\
&\leq \left(\|\mathbf{g}_n - \mathbf{g}^*\|_v^2 - 2\gamma_{n+1}\langle \nabla_{\mathbf{g}} h(\mathbf{g}_n, X_{n+1}), \mathbf{g}_n - \mathbf{g}^* \rangle_v + 4\gamma_{n+1}^2\right) .
\end{aligned} \tag{14}$$

Using Proposition B.16, we have for any $\mathbf{g} \in \mathcal{C}$, $\langle \nabla H(\mathbf{g}_n), \mathbf{g}_n - \mathbf{g}^* \rangle_v \geq \eta \|\mathbf{g}_n - \mathbf{g}^*\|_v^2$. Therefore, taking the conditional expectation, we obtain

$$\mathbb{E}\left[\|\mathbf{g}_{n+1} - \mathbf{g}^*\|_v^2 \mid \mathcal{F}_n\right] \leq \|\mathbf{g}_n - \mathbf{g}^*\|_v^2(1 - 2\eta\gamma_{n+1}) + 4\gamma_{n+1}^2 .$$

Taking the expectation and applying Lemma F.3 with $\delta_n = \mathbb{E}[\|\mathbf{g}_n - \mathbf{g}^*\|_v^2]$ and $m_n = 4\gamma_n$ gives

$$\mathbb{E}\left[\|\mathbf{g}_{n+1} - \mathbf{g}^*\|_v^2\right] \leq \exp\left(-2\eta \sum_{k=\lceil n/2 \rceil}^{n} \gamma_k\right)\left(\sum_{n=n_0}^{n} 4\gamma_n^2 + \mathbb{E}[\|\mathbf{g}_{n_0} - \mathbf{g}^*\|_v^2]\right) + \frac{4}{\eta}\gamma_{\lceil n/2 \rceil - 1}$$

where $n_0 = \min\{n \in \mathbb{N}, \eta\gamma_{n+1} \leq 1\}$. Remark that the exponential term converges exponentially fast. Indeed, we have $\sum_{k=\lceil n/2 \rceil}^{n} \gamma_k \gtrsim n^{1-b}$ with $1 - b > 0$. Moreover, $\|\mathbf{g}_{n_0} - \mathbf{g}^*\|_v^2 \leq \mathrm{Diam}_2(\mathcal{C})^2$. Therefore, since for all $n \geq 2$, $\gamma_{\lceil n/2 \rceil - 1} \leq 2^b\gamma_n$, we have the desired convergence rate

$$\mathbb{E}\left[\|\mathbf{g}_{n+1} - \mathbf{g}^*\|_v^2\right] \lesssim \frac{\gamma_n}{\eta}$$

which concludes the proof.

□

#### C.1.2  Convergence rate for higher order moments of the non-averaged iterates.

We prove here the convergence rate of higher order moments of the error $\|\mathbf{g}_n - \mathbf{g}^*\|_v$. This convergence will be useful for the convergence rate of the averaged iterates of PSGD. While this proposition directly proves Theorem 4.4 by the use of Jensen's inequality, the proof is slightly more cumbersome so we decided to make a separate case.

**Proposition C.1.** *Under Assumptions (A1) or (B1-3) and for any decay step of the form $\gamma_n = \gamma_1/n^b$ with $\gamma_1 > 0, b \in (1/2, 1)$ and $p \in \{1, 2, 3\}$, we have the convergence rate*

$$\mathbb{E}\left[\|\mathbf{g}_n - \mathbf{g}^*\|^{2p}\right] \lesssim \frac{\gamma_n^p}{\eta^p}$$

*where $(\mathbf{g}_n)$ is the sequence of non-averaged iterates of PSGD.*

*Proof.* By definition of the gradient step at time $n + 1$, sampling $X_{n+1} \sim \mu$ and using inequality (14),

$$\|\mathbf{g}_{n+1} - \mathbf{g}^*\|_v^6 \leq \left(\|\mathbf{g}_n - \mathbf{g}^*\|_v^2 - 2\gamma_{n+1}\langle\nabla_\mathbf{g} h(\mathbf{g}_n, X_{n+1}), \mathbf{g}_n - \mathbf{g}^*\rangle_v + 4\gamma_{n+1}^2\right)^3 .$$

Using that $(A + B + C)^3 = \sum_{a+b+c=3} \frac{3!}{a!b!c!} A^a B^b C^c$, we obtain

$$\begin{aligned}
\|\mathbf{g}_{n+1} - \mathbf{g}^*\|_v^6 \leq\ & \|\mathbf{g}_n - \mathbf{g}^*\|_v^6 \\
& - 6\|\mathbf{g}_n - \mathbf{g}^*\|_v^4 \gamma_{n+1}\langle\nabla_\mathbf{g} h(\mathbf{g}_n, X_{n+1}), \mathbf{g}_n - \mathbf{g}^*\rangle_v \\
& + 12\|\mathbf{g}_n - \mathbf{g}^*\|_v^4 \gamma_{n+1}^2 \\
& + 12\|\mathbf{g}_n - \mathbf{g}^*\|_v^2 \gamma_{n+1}^2 \langle\nabla_\mathbf{g} h(\mathbf{g}_n, X_{n+1}), \mathbf{g}_n - \mathbf{g}^*\rangle_v^2 \\
& - 48\|\mathbf{g}_n - \mathbf{g}^*\|_v^2 \gamma_{n+1}^3 \langle\nabla_\mathbf{g} h(\mathbf{g}_n, X_{n+1}), \mathbf{g}_n - \mathbf{g}^*\rangle_v \\
& + 48\|\mathbf{g}_n - \mathbf{g}^*\|_v^2 \gamma_{n+1}^4 \\
& - 8\gamma_{n+1}^3 \langle\nabla_\mathbf{g} h(\mathbf{g}_n, X_{n+1}), \mathbf{g}_n - \mathbf{g}^*\rangle_v^3 \\
& + 48\gamma_{n+1}^4 \langle\nabla_\mathbf{g} h(\mathbf{g}_n, X_{n+1}), \mathbf{g}_n - \mathbf{g}^*\rangle_v^2 \\
& - 96\gamma_{n+1}^5 \langle\nabla_\mathbf{g} h(\mathbf{g}_n, X_{n+1}), \mathbf{g}_n - \mathbf{g}^*\rangle_v \\
& + 2^6\gamma_{n+1}^6.
\end{aligned}$$

Taking the conditional expectation and already omitting some negative terms thanks to $\langle\nabla H(\mathbf{g}_n), \mathbf{g}_n - \mathbf{g}^*\rangle_v \geq 0$, which follows from the fact that $H$ is convex and $\mathbf{g}^*$ is a minimizer, gives the simplification

$$\begin{aligned}
\mathbb{E}\left[\|\mathbf{g}_{n+1} - \mathbf{g}^*\|_v^6 \mid \mathcal{F}_n\right] \leq\ & \|\mathbf{g}_n - \mathbf{g}^*\|_v^6 \\
& - 6\|\mathbf{g}_n - \mathbf{g}^*\|_v^4 \gamma_{n+1}\langle\nabla H(\mathbf{g}_n), \mathbf{g}_n - \mathbf{g}^*\rangle_v \\
& + 12\|\mathbf{g}_n - \mathbf{g}^*\|_v^4 \gamma_{n+1}^2 \\
& + 12\|\mathbf{g}_n - \mathbf{g}^*\|_v^2 \gamma_{n+1}^2 \mathbb{E}\left[\langle\nabla h(\mathbf{g}_n, X_{n+1}), \mathbf{g}_n - \mathbf{g}^*\rangle_v^2 \mid \mathcal{F}_n\right] \\
& + 48\|\mathbf{g}_n - \mathbf{g}^*\|_v^2 \gamma_{n+1}^4 \\
& + 48\gamma_{n+1}^4 \mathbb{E}\left[\langle\nabla h(\mathbf{g}_n, X_{n+1}), \mathbf{g}_n - \mathbf{g}^*\rangle_v^2 \mid \mathcal{F}_n\right] \\
& + 2^6\gamma_{n+1}^6 \\
& - 8\gamma_{n+1}^3 \mathbb{E}\left[\langle\nabla_\mathbf{g} h(\mathbf{g}_n, X_{n+1}), \mathbf{g}_n - \mathbf{g}^*\rangle_v^3 \mid \mathcal{F}_n\right] .
\end{aligned}$$

Using Proposition B.16, we have for any $\mathbf{g} \in \mathcal{C}$, $\langle\nabla H(\mathbf{g}_n), \mathbf{g}_n - \mathbf{g}^*\rangle_v \geq \eta\|\mathbf{g}_n - \mathbf{g}^*\|_v^2$. The Cauchy-Schwarz inequality gives $|\langle\nabla h(\mathbf{g}_n, X_{n+1}), \mathbf{g}_n - \mathbf{g}^*\rangle_v| \leq 2\|\mathbf{g}_n - \mathbf{g}^*\|_v$. Combining these two inequalities we obtain

$$\begin{aligned}
\mathbb{E}\left[\|\mathbf{g}_{n+1} - \mathbf{g}^*\|_v^6 \mid \mathcal{F}_n\right] \leq\ & \|\mathbf{g}_n - \mathbf{g}^*\|_v^6(1 - 6\eta\gamma_{n+1}) + 60\gamma_{n+1}^2\|\mathbf{g}_n - \mathbf{g}^*\|_v^4 \\
& + 240\gamma_{n+1}^4\|\mathbf{g}_n - \mathbf{g}^*\|_v^2 \quad + 2^6\gamma_{n+1}^6 + 64\gamma_{n+1}^3\|\mathbf{g}_n - \mathbf{g}^*\|_v^3 .
\end{aligned}$$

Using Young's (generalized) inequality $ab = ac\frac{b}{c} \leq \frac{(ac)^p}{p} + \frac{b^q}{c^q q}$ for $c \neq 0, \frac{1}{p} + \frac{1}{q} = 1$ and applying it to $60\gamma_{n+1}\|\mathbf{g}_n - \mathbf{g}^*\|_v^4$ with $c = \left(\frac{2}{3\eta}\right)^{2/3}, p = 3, q = \frac{3}{2}$ gives $60\gamma_{n+1}\|\mathbf{g}_n - \mathbf{g}^*\|_v^4 \leq \frac{60^3\gamma_{n+1}^3}{3}$ .

$(\frac{2}{3\eta})^2 + \eta\|\mathbf{g}_n - \mathbf{g}^*\|_v^6$. Analogously, one has $64\gamma_{n+1}^2\|\mathbf{g}_n - \mathbf{g}^*\|_v^3 \leq \frac{2^{10}}{\eta}\gamma_{n+1}^4 + \eta\|\mathbf{g}_n - \mathbf{g}^*\|_v^6$. Thus, taking the expectation, we have

$$\mathbb{E}[\|\mathbf{g}_{n+1} - \mathbf{g}^*\|_v^6] \leq \mathbb{E}[\|\mathbf{g}_n - \mathbf{g}^*\|_v^6](1 - 4\eta\gamma_{n+1}) + \gamma_{n+1}^4\left(240\cdot\mathrm{Diam}_2(\mathcal{C})^2 + \frac{32}{\eta^2}\cdot 10^3\right)$$

$$+ \frac{2^{10}}{\eta}\gamma_{n+1}^5 + 64\gamma_{n+1}^6,$$

where the terms involving $\mathrm{Diam}_2(\mathcal{C})$ appears from the crude bound $\|\mathbf{g}_n - \mathbf{g}\|_v \leq \mathrm{Diam}_2(\mathcal{C})$.

Applying Lemma F.3 in a similar way as in the proof of Theorem 4.4 gives $\mathbb{E}[\|\mathbf{g}_{n+1} - \mathbf{g}^*\|_v^6] \lesssim \frac{\gamma_n^3}{\eta^3}$, so by Jensen's inequality, we conclude

$$\mathbb{E}[\|\mathbf{g}_{n+1} - \mathbf{g}^*\|_v^{2p}] \lesssim \frac{\gamma_n^p}{\eta^p} \quad \text{for } p \in \{1, 2, 3\}\,.$$

$\square$

### C.2 Convergence of the averaged iterates.

**Theorem** 4.5 (Averaged iterates) Under Assumptions (A1) or (B1–B3), and assuming that $f_\mu$ is $\alpha$-Hölder with $\alpha \in (0, 1]$, for any decay schedule of the form $\gamma_n = \gamma_1/n^b$ with $\gamma_1 > 0$ and $b \in \left(\frac{1}{1+\alpha}, 1\right)$, we have the convergence rate

$$\mathbb{E}[\|\bar{\mathbf{g}}_n - \mathbf{g}^*\|_v^2] = \mathcal{O}\left(1/n\right).$$

Without assuming $f_\mu$ to be $\alpha$-Hölder, and for $b \in (1/2, 1)$, we still obtain

$$\mathbb{E}[\|\bar{\mathbf{g}}_n - \mathbf{g}^*\|_v^2] = \mathcal{O}\left(1/n^b\right).$$

*Proof.* For this proof, we introduce the additional following notation:

For any $c > 0$ we define the function $t \mapsto \Psi_c(t)$ such that

$$\sum_{t=1}^T t^{-c} \leq \Psi_c(T) := \begin{cases} 1 + \ln(T+1) & \text{if } c = 1, \\ \frac{2c-1}{c-1} & \text{if } c > 1, \\ 1 + \frac{1}{1-c}(T+1)^{1-c} & \text{if } c < 1. \end{cases} \quad (15)$$

We start by a decomposition of the gradient step, already present in [19]. We define the differences

$$\mathbf{p}_k := \mathrm{Proj}_{\mathcal{C}}\left(\mathbf{g}_k - \gamma_{k+1}\nabla_{\mathbf{g}}h\left(\mathbf{g}_k, X_{k+1}\right)\right) - \left(\mathbf{g}_k - \gamma_{k+1}\nabla_{\mathbf{g}}h\left(\mathbf{g}_k, X_{k+1}\right)\right),$$
$$\boldsymbol{\xi}_{k+1} := \nabla H\left(\mathbf{g}_k\right) - \nabla_{\mathbf{g}}h\left(\mathbf{g}_k, X_{k+1}\right),$$
$$\boldsymbol{\delta}_k := \nabla H\left(\mathbf{g}_k\right) - \nabla^2 H(\mathbf{g}^*)\left(\mathbf{g}_k - \mathbf{g}^*\right).$$

Noting $I_M$ the identity matrix in $\mathcal{M}_M(\mathbb{R})$, we observe that, by incorporating each introduced term sequentially, for any $k \in \mathbb{N}$, we have

$$\begin{aligned}
\mathbf{g}_{k+1} - \mathbf{g}^* &= \mathrm{Proj}_{\mathcal{C}}\left(\mathbf{g}_k - \gamma_{k+1}\nabla_{\mathbf{g}}h(\mathbf{g}_k, X_{k+1})\right) - \mathbf{g}^* \\
&= \mathbf{g}_k - \gamma_{k+1}\nabla_{\mathbf{g}}h(\mathbf{g}_k, X_{k+1}) - \mathbf{g}^* - \mathbf{p}_k \\
&= \mathbf{g}_k - \gamma_{k+1}\nabla H(\mathbf{g}_k, X_{k+1}) - \mathbf{g}^* + \gamma_{k+1}\boldsymbol{\xi}_{k+1} - \mathbf{p}_k \\
&= \left(I_M - \gamma_{k+1}\nabla^2 H(\mathbf{g}^*)\right)\left(\mathbf{g}_k - \mathbf{g}_*\right) - \gamma_{k+1}\boldsymbol{\delta}_k + \gamma_{k+1}\boldsymbol{\xi}_{k+1} + \mathbf{p}_k\,.
\end{aligned}$$

Thus, we have that

$$\nabla^2 H(\mathbf{g}^*)\left(\mathbf{g}_k - \mathbf{g}^*\right) = \frac{\mathbf{g}_k - \mathbf{g}_{k+1}}{\gamma_{k+1}} - \boldsymbol{\delta}_k + \boldsymbol{\xi}_{k+1} + \frac{\mathbf{p}_k}{\gamma_{k+1}}\,.$$

Observing that $\frac{1}{n+1}\sum_{k=0}^n(\mathbf{g}_k - \mathbf{g}^*) = \bar{\mathbf{g}}_n - \mathbf{g}^*$, we have the following decomposition of the averaged iterates

$$\nabla^2 H(\mathbf{g}^*)(\bar{\mathbf{g}}_n - \mathbf{g}^*) = \frac{1}{n+1}\sum_{k=0}^n \frac{\mathbf{g}_k - \mathbf{g}_{k+1}}{\gamma_{k+1}} - \frac{1}{n+1}\sum_{k=0}^n \boldsymbol{\delta}_k + \frac{1}{n+1}\sum_{k=0}^n \boldsymbol{\xi}_{k+1} + \frac{1}{n+1}\sum_{k=0}^n \frac{\mathbf{p}_k}{\gamma_{k+1}}\,.$$

We will now give the convergence rate of each sum.

- **Convergence rate for $\frac{1}{n+1}\sum_{k=0}^{n}\frac{\mathbf{g}_k-\mathbf{g}_{k+1}}{\gamma_{k+1}}$.**

$$\sum_{k=0}^{n}\frac{\mathbf{g}_k-\mathbf{g}_{k+1}}{\gamma_{k+1}}=\sum_{k=0}^{n}\frac{(\mathbf{g}_k-\mathbf{g}^*)-(\mathbf{g}_{k+1}-\mathbf{g}^*)}{\gamma_{k+1}}$$

$$=\sum_{k=0}^{n}\frac{\mathbf{g}_k-\mathbf{g}^*}{\gamma_{k+1}}-\sum_{k=0}^{n}\frac{\mathbf{g}_{k+1}-\mathbf{g}^*}{\gamma_{k+1}}$$

$$=\sum_{k=1}^{n}\left(\frac{1}{\gamma_{k+1}}-\frac{1}{\gamma_k}\right)(\mathbf{g}_k-\mathbf{g}^*)+\frac{\mathbf{g}_0-\mathbf{g}^*}{\gamma_1}-\frac{\mathbf{g}_{n+1}-\mathbf{g}^*}{\gamma_{n+1}}.$$

Remark that $\gamma_{n+1}^{-1}-\gamma_n^{-1}\leq 2\gamma_1^{-1}n^{b-1}$. Using Minkowski's inequality and that, by Theorem 4.4 (non-averaged iterates), $\mathbb{E}\left[\|\mathbf{g}_n-\mathbf{g}^*\|_v^2\right]\lesssim\frac{\gamma_1}{\eta}(n+1)^{-b}$,

$$\mathbb{E}\left[\left\|\sum_{k=0}^{n}\frac{\mathbf{g}_k-\mathbf{g}_{k+1}}{\gamma_{k+1}}\right\|_v^2\right]^{\frac{1}{2}}\lesssim\frac{1}{\eta}\Psi_{1-b/2}(n+1)+\mathrm{Diam}_2(\mathcal{C})\gamma_1^{-1}+\frac{1}{\sqrt{\gamma_1\eta}}(n+1)^{b/2}\,.$$

We thus have the convergence rate

$$\frac{1}{n+1}\mathbb{E}\left[\left\|\sum_{k=0}^{n}\frac{\mathbf{g}_k-\mathbf{g}_{k+1}}{\gamma_{k+1}}\right\|_v^2\right]^{\frac{1}{2}}\lesssim\frac{1}{\eta(n+1)^{1-b/2}}\,.$$

- **Convergence rate for $\frac{1}{n+1}\sum_{k=0}^{n}\boldsymbol{\delta}_k$.**

We recall that $\boldsymbol{\delta}_k=\nabla H(\mathbf{g}_k)-\nabla^2 H(\mathbf{g}^*)(\mathbf{g}_k-\mathbf{g}^*)$ and that the Hessian using either Theorem B.3 or Proposition 4.1, depending on our setting, there exists a ball $B(\mathbf{g}^*,d_1)$ with $d_1>0$ where $H$ is $\alpha$-Hölder. Therefore, applying Lemma F.4, if $\mathbf{g}_k\in B(\mathbf{g}^*,d_1)$, we have

$$\|\boldsymbol{\delta}_k\|\lesssim\|\mathbf{g}_k-\mathbf{g}^*\|_v^{1+\alpha}\,.$$

Otherwise, since the Hessian, whose expression is provided in Proposition B.5, is uniformly bounded by an application of Lemma F.2, there exists a constant $C_{\boldsymbol{\delta}}$ such that for any $\mathbf{g}\in\mathcal{C}$, $\|\nabla H(\mathbf{g})-\nabla^2 H(\mathbf{g}^*)(\mathbf{g}-\mathbf{g}^*)\|\leq C_{\boldsymbol{\delta}}$.

Since $\mathbb{P}(\mathbf{g}_k\notin B(\mathbf{g}^*,d_1))=\mathbb{P}(\|\mathbf{g}_k-\mathbf{g}_*\|>d_1)$, using Markov's inequality gives

$$\mathbb{E}[\|\boldsymbol{\delta}_k\|_v^2]=\mathbb{E}[\|\boldsymbol{\delta}_k\|_v^2\mathbf{1}_{\mathbf{g}_k\in B(\mathbf{g}^*,d_1)}]+\mathbb{E}[\|\boldsymbol{\delta}_k\|_v^2\mathbf{1}_{\mathbf{g}_k\notin B(\mathbf{g}^*,d_1)}]$$

$$\lesssim\mathbb{E}\left[\|\mathbf{g}_k-\mathbf{g}^*\|_v^{2+2\alpha}\right]+\frac{C_{\boldsymbol{\delta}}^2}{d_1^{2+2\alpha}}\mathbb{E}[\|\mathbf{g}_k-\mathbf{g}^*\|_v^{2+2\alpha}]$$

$$\lesssim\mathbb{E}[\|\mathbf{g}_k-\mathbf{g}^*\|_v^{2+2\alpha}]\,.$$

Therefore, using Minkowski's inequality, we have

$$\frac{1}{n+1}\mathbb{E}\left[\left\|\sum_{k=0}^{n}\boldsymbol{\delta}_k\right\|_v^2\right]^{\frac{1}{2}}\lesssim\frac{1}{n+1}\sum_{k=0}^{n}\frac{1}{\eta^{\frac{1+\alpha}{2}}}\gamma_{k+1}^{\frac{1+\alpha}{2}}$$

$$\leq\frac{1}{\eta^{\frac{1+\alpha}{2}}(n+1)}\Psi_{\frac{b+\alpha b}{2}}$$

$$\lesssim\frac{1}{\eta^{\frac{1+\alpha}{2}}(n+1)^{\frac{b+\alpha b}{2}}}\,.$$

- **Convergence rate for $\frac{1}{n+1}\sum_{k=0}^{n}\boldsymbol{\xi}_{k+1}$.**

We recall that $\boldsymbol{\xi}_{k+1} = \nabla H\left(\mathbf{g}_k\right) - \nabla_{\mathbf{g}} h\left(\mathbf{g}_k, X_{k+1}\right)$ and thus $\mathbb{E}[\boldsymbol{\xi}_{k+1}] = 0$.

Observe that

$$\mathbb{E}\left[\left\|\sum_{k=0}^{n}\boldsymbol{\xi}_{k+1}\right\|_v^2\right] = \mathbb{E}\left[\left\|\sum_{k=0}^{n-1}\boldsymbol{\xi}_{k+1}\right\|_v^2 + 2\left\langle\sum_{k=0}^{n-1}\boldsymbol{\xi}_{k+1}, \boldsymbol{\xi}_{n+1}\right\rangle_v + \|\boldsymbol{\xi}_{n+1}\|_v^2\right]$$

with

$$\mathbb{E}\left[\left\langle\sum_{k=0}^{n-1}\boldsymbol{\xi}_{k+1}, \boldsymbol{\xi}_{n+1}\right\rangle_v\right] = \mathbb{E}\left[\left\langle\sum_{k=0}^{n-1}\boldsymbol{\xi}_{k+1}, \mathbb{E}\left[\boldsymbol{\xi}_{n+1}|\mathcal{F}_n\right]\right\rangle_v\right] = 0 \ .$$

Thus, since for all $k$, $\mathbb{E}\left[\|\boldsymbol{\xi}_k\|^2\right] \le 4$, we have the convergence rate

$$\frac{1}{n+1}\mathbb{E}\left[\left\|\sum_{k=0}^{n}\boldsymbol{\xi}_{k+1}\right\|_v^2\right]^{\frac{1}{2}} \le \frac{2}{\sqrt{n+1}} \ .$$

- **Convergence rate for** $\frac{1}{n+1}\sum_{k=0}^{n}\frac{\mathbf{p}_k}{\gamma_{k+1}}$.

Take $d_0$ such that $B(\mathbf{g}^*, d_0) \subset \mathcal{C}$. Using the notation $\nabla_k := \nabla_{\mathbf{g}} h\left(\mathbf{g}_k, X_{k+1}\right)$ for conciseness, we obtain

$$\mathbb{E}\left[\|\mathbf{p}_k\|_v^2\right] = \mathbb{E}\left[\left\|\text{Proj}_{\mathcal{C}}\left(\mathbf{g}_k - \gamma_{k+1}\nabla_k\right) - \left(\mathbf{g}_k - \gamma_{k+1}\nabla_k\right)\right\|_v^2\right]$$
$$= \mathbb{E}\left[\left\|\text{Proj}_{\mathcal{C}}\left(\mathbf{g}_k - \gamma_{k+1}\nabla_k\right) - \left(\mathbf{g}_k - \gamma_{k+1}\nabla_k\right)\right\|_v^2 \mathbf{1}_{\mathbf{g}_k - \gamma_{k+1}\nabla_k \notin \mathcal{C}}\right]$$

Since for any $y \in \mathcal{C}$, one has $\|x - \text{Proj}_{\mathcal{C}}(x)\|_v \le \|x - y\|_v$, taking $y = \mathbf{g_k}$, and since $\mathbf{g}_k - \gamma_{k+1}\nabla_k \notin \mathcal{C}$ is satisfied only if $\|\mathbf{g}_k - \gamma_{k+1}\nabla_k - \mathbf{g}^*\|_v > d_0$, we have

$$\mathbb{E}\left[\|\mathbf{p}_k\|_v^2\right] \le \mathbb{E}\left[\|\gamma_{k+1}\nabla_k\|_v^2 \mathbf{1}_{\|\mathbf{g}_k - \gamma_{k+1}\nabla_k - \mathbf{g}^*\|_v > d_0}\right]$$
$$\le 4\gamma_{k+1}^2 \frac{\mathbb{E}\left[\|\mathbf{g}_k - \gamma_{k+1}\nabla_k - \mathbf{g}^*\|_v^4\right]}{d_0^4}$$
$$\le \frac{\gamma_{k+1}^2}{d_0^4}\left(2^5\mathbb{E}\left[\|\mathbf{g}_k - \mathbf{g}^*\|_v^4\right] + 2^9\gamma_{k+1}^4\right)$$
$$\lesssim \frac{1}{\eta^2}\gamma_{k+1}^4$$

Where we used Markov's inequality and the inequality $(A + B)^4 \le 2^3(A^4 + B^4)$, for all $A, B \in \mathbb{R}$ and that by Proposition C.1, $\mathbb{E}\left[\|\mathbf{g}_n - \mathbf{g}^*\|^4\right] \lesssim \frac{\gamma_1^2}{\eta^2}(n+1)^{-2b}$.

We thus have

$$\frac{1}{n+1}\mathbb{E}\left[\left\|\sum_{k=0}^{n}\frac{\mathbf{p}_k}{\gamma_{k+1}}\right\|_v^2\right]^{\frac{1}{2}} \lesssim \frac{1}{n+1}\sum_{k=0}^{n}\frac{\gamma_{k+1}}{\eta}$$
$$\lesssim \frac{\gamma_1}{\eta(n+1)^b} \ .$$

- **Conclusion.**

Using the convergence rate of all our terms, Cauchy-Schwarz inequality and that $(A + B)^2 \leq 2(A^2 + B^2)$ for all $A, B \in \mathbb{R}$ we conclude

$$\mathbb{E}\left[\left\|\nabla^2 H(\mathbf{g}^*)(\overline{\mathbf{g}}_n - \mathbf{g}^*)\right\|_v^2\right]$$

$$= \mathbb{E}\left[\left\|\frac{1}{n+1}\sum_{k=0}^{n}\frac{\mathbf{g}_k - \mathbf{g}_{k+1}}{\gamma_{k+1}} - \frac{1}{n+1}\sum_{k=0}^{n}\boldsymbol{\delta}_k + \frac{1}{n+1}\sum_{k=0}^{n}\boldsymbol{\xi}_{k+1} + \frac{1}{n+1}\sum_{k=0}^{n}\frac{\mathbf{p}_k}{\gamma_{k+1}}\right\|_v^2\right]$$

$$\lesssim \frac{1}{\eta^2(n+1)^{2-b}} + \frac{1}{\eta^{\frac{1+\alpha}{\alpha}}(n+1)^{b+\alpha b}} + \frac{1}{n+1} + \frac{\gamma_1^4}{\eta^2(n+1)^{2b}}.$$

Since $b \in (\frac{1}{2}, 1)$ and $b \in (\frac{1}{1+\alpha}, 1)$ the limiting term is $\frac{1}{n+1}$ and we have

$$\mathbb{E}\left[\left\|\nabla^2 H(\mathbf{g}^*)(\overline{\mathbf{g}}_n - \mathbf{g}^*)\right\|_v^2\right] \lesssim \frac{1}{n+1}.$$

Finally, observe that there is an orthogonal matrix $U$ such that

$$\nabla^2 H(\mathbf{g}^*) = U \operatorname{diag}(\lambda_1, \ldots, \lambda_{M-1}, 0) U^\top.$$

. Therefore, noting by abuse of notation

$$\left(\nabla^2 H(\mathbf{g}^*)\right)^{-1} = U \operatorname{diag}\left(\lambda_1^{-1}, \ldots, \lambda_{M-1}^{-1}, 0\right) U^\top$$

the inverse of $\nabla_k^2$ in the space $\operatorname{Vect}(\mathbf{1}_M)^\perp$ we finally have

$$\mathbb{E}[\|\overline{\mathbf{g}}_n - \mathbf{g}^*\|_v^2] \lesssim \frac{1}{\lambda^2(n+1)}$$

where $\lambda = \min_{j \in [\![1, M-1]\!]} \lambda_j > 0$ by either Theorem B.3 or Proposition B.12. $\qquad\square$

## C.3 Convergence rate of PSGD in the general setting

**Theorem** 3.2 (PSGD in the general setting) Assuming that the semi-dual problem (4) admits at least one solution $\mathbf{g}^*$ and that there exists a compact set $K$ such that $\mu(K) \geq 1 - \frac{1}{2}w_{\min}$, choosing the learning rate $\gamma_n = \gamma_1/n^b$ with $\gamma_1 = \frac{\operatorname{Diam}(\mathcal{C})}{2\sqrt{2}}$ and $b = \frac{1}{2}$, we obtain

$$\mathbb{E}[H(\overline{\mathbf{g}}_k) - H(\mathbf{g}^*)] \leq \frac{4\sqrt{2}\operatorname{Diam}(\mathcal{C})}{\sqrt{n}}.$$

*Proof.* Define $\gamma_k = \frac{\gamma_1}{\sqrt{k}}$ for $\gamma_1 > 0, k \geq 1$ and denote by $\mathbf{g}^* \in \mathcal{C}$ a minimizer of the functional $H$. Thanks to Jensen's inequality coupled with the fact that no matter $\mathbf{g} \in \mathbb{R}^M : H(\mathbf{g}) - H(\mathbf{g}^*) \leq \nabla H(\mathbf{g})^\top (\mathbf{g} - \mathbf{g}^*)$, it comes

$$\mathbb{E}[H(\overline{\mathbf{g}}_n) - H(\mathbf{g}^*)] \leq \mathbb{E}\left[\frac{1}{n+1}\sum_{k=0}^{n}H(\mathbf{g}_k) - H(\mathbf{g}^*)\right] \leq \frac{1}{n+1}\mathbb{E}\left[\sum_{k=0}^{n}\nabla H(\mathbf{g}_k)^\top(\mathbf{g}_k - \mathbf{g}^*)\right].$$

Then, since no matter $k$, $X_{k+1} \sim \mu$, we have $\mathbb{E}[\nabla_{\mathbf{g}}h(\mathbf{g}_k, X_{k+1}) \mid \mathcal{F}_k] = \nabla H(\mathbf{g}_k)$, we have

$$\mathbb{E}[H(\overline{\mathbf{g}}_n) - H(\mathbf{g}^*)] \leq \frac{1}{n+1}\mathbb{E}\left[\sum_{k=0}^{n}\nabla_{\mathbf{g}}h(\mathbf{g}_k, X_{k+1})^\top(\mathbf{g}_k - \mathbf{g}^*)\right]. \tag{16}$$

We will proceed to bound the right hand side of this inequality.

By definition of $\mathbf{g}_{k+1}$, and since $\mathbf{g}^* \in \mathcal{C}$ and $\operatorname{Proj}_{\mathcal{C}}$ is 1-Lipschitz, it comes

$$\mathbb{E}\left[\|\mathbf{g}_{k+1} - \mathbf{g}^*\|^2\right] \leq \mathbb{E}\left[\|\operatorname{Proj}_{\mathcal{C}}(\mathbf{g}_k - \gamma_{k+1}\nabla_{\mathbf{g}}h(\mathbf{g}_k, X_{k+1})) - \mathbf{g}^*\|^2\right]$$

$$\leq \mathbb{E}\left[\|\mathbf{g}_k - \gamma_{k+1}\nabla_{\mathbf{g}}h(\mathbf{g}_k, X_{k+1}) - \mathbf{g}^*\|^2\right]$$

$$\leq \mathbb{E}\left[\|\mathbf{g}_k - \mathbf{g}^*\|^2 + \gamma_{k+1}^2\|\nabla_{\mathbf{g}}h(\mathbf{g}_k, X_{k+1})\|^2 - 2\gamma_{k+1}\nabla_{\mathbf{g}}h(\mathbf{g}_k, X_{k+1})^\top(\mathbf{g}_k - \mathbf{g}^*)\right].$$

In addition, since $\|\nabla_{\mathbf{g}} h(\mathbf{g}, X)\| \leq 2$ a.s. no matter $\mathbf{g} \in \mathbb{R}^M$ and $X \in \mathbb{R}^d$,

$$\mathbb{E}\left[2\gamma_{k+1}\nabla_{\mathbf{g}}h\left(\mathbf{g}_k, X_{k+1}\right)^{\top}\left(\mathbf{g}_k - \mathbf{g}^*\right)\right] \leq \mathbb{E}\left[\|\mathbf{g}_k - \mathbf{g}^*\|^2 - \|\mathbf{g}_{k+1} - \mathbf{g}^*\|^2 + 4\gamma_{k+1}^2\right].$$

Then, with the help of Abel's summation formula,

$$2\mathbb{E}\left[\sum_{k=0}^{n}\nabla_{\mathbf{g}}h\left(\mathbf{g}_k, X_{k+1}\right)^{\top}\left(\mathbf{g}_k - \mathbf{g}^*\right)\right]$$

$$\leq \mathbb{E}\left[\sum_{k=0}^{n}\frac{\|\mathbf{g}_k - \mathbf{g}^*\|^2 - \|\mathbf{g}_{k+1} - \mathbf{g}^*\|^2}{\gamma_{k+1}}\right] + 4\sum_{k=0}^{n}\gamma_{k+1}$$

$$\leq \mathbb{E}\left[\sum_{k=1}^{n}\|\mathbf{g}_k - \mathbf{g}^*\|^2\left(\frac{1}{\gamma_{k+1}} - \frac{1}{\gamma_k}\right)\right] - \frac{\|\mathbf{g}_{k+1} - \mathbf{g}^*\|^2}{\gamma_{k+1}} + \frac{\|\mathbf{g}_0 - \mathbf{g}^*\|^2}{\gamma_1} + 4\sum_{k=0}^{n}\gamma_{k+1} .$$

Then, since for all $k$, $\|\mathbf{g}_k - \mathbf{g}^*\| \leq \mathrm{Diam}_2(\mathcal{C})$, it comes

$$2\mathbb{E}\left[\sum_{k=0}^{n}\nabla_{\mathbf{g}}h\left(\mathbf{g}_k, X_{k+1}\right)^{\top}\left(\mathbf{g}_k - \mathbf{g}^*\right)\right] \leq \mathrm{Diam}_2(\mathcal{C})^2\sum_{k=1}^{n}\left(\frac{1}{\gamma_{k+1}} - \frac{1}{\gamma_k}\right) + \frac{D^2}{\gamma_1} + 4\sum_{k=1}^{n}\gamma_k$$

$$\leq \frac{\mathrm{Diam}_2(\mathcal{C})^2}{\gamma_{n+1}} + 4\sum_{k=1}^{n}\gamma_k$$

$$\leq \frac{\mathrm{Diam}_2(\mathcal{C})^2}{\gamma_1}\sqrt{n+1} + 8\gamma_1\sqrt{n+1}$$

Using the inequality (16) we obtain

$$\mathbb{E}\left[H\left(\overline{\mathbf{g}}_n\right) - H\left(\mathbf{g}^*\right)\right] \leq \frac{1}{\sqrt{n+1}}\left(\frac{\mathrm{Diam}_2(\mathcal{C})^2}{\gamma_1} + 8\gamma_1\right).$$

The best constant $\gamma_1$ is then $\gamma_1 = \frac{\mathrm{Diam}_2(\mathcal{C})}{2\sqrt{2}}$, but no matter $\gamma_1 > 0$, we have the desired convergence rate

$$\mathbb{E}\left[H\left(\overline{\mathbf{g}}_n\right) - H\left(\mathbf{g}^*\right)\right] = \mathcal{O}(1/\sqrt{n}).$$

$\square$

## D Proof of Lemma 3.1: Localisation of a projection set

**Lemma** 3.1 (Existence of a projection set) As soon as the semi-dual problem is well-posed, there exists a minimizer $\mathbf{g}^*$ in the set

$$\mathcal{C} := \left\{\mathbf{g} \in \{0\} \times \mathbb{R}^{M-1} \mid |g_j| \leq \|c\|_{K,\infty}\right\}$$

where $\|c\|_{K,\infty} := \sup_{x \in K, j \in [\![1,M]\!]} |c(x, y_j)|$ for any compact set $K$ satisfying $\mu(K) \geq 1 - \frac{1}{2}\min w_j$.

*Proof.* By the first order condition, the minimizer of $H$ satisfies $\mu(\mathbb{L}_j(\mathbf{g})) = w_j$ for all $j \in [\![1, M]\!]$. In particular, one can restrict the search set for an optimal potential to the set of potentials defined by

$$\mathcal{L} := \left\{\mathbf{g} \in \mathbb{R}^M : \forall j \in [\![1, M]\!], \mu(\mathbb{L}_j(\mathbf{g})) \geq \frac{2}{3}w_{\min}\right\}.$$

Let us show that this set is contained in an $L^\infty$ ball with an explicit radius. Consider any compact set $K$ such that $\mu(K) \geq 1 - \frac{1}{2}w_{\min}$. For $\mathbf{g} \in \mathcal{L}$ and any $j \in [\![1, M]\!]$ we get that

$$\mu(\mathbb{L}_j(\mathbf{g}) \cap K) = 1 - \mu((\mathbb{R}^M \setminus \mathbb{L}_j(\mathbf{g})) \cup (\mathbb{R}^M \setminus K))$$

$$\geq 1 - \mu((\mathbb{R}^M \setminus \mathbb{L}_j(\mathbf{g}))) - \mu((\mathbb{R}^M \setminus K))$$

$$= 1 - (1 - \mu(\mathbb{L}_j(\mathbf{g}))) - (1 - \mu(K))$$

$$\geq \frac{w_{\min}}{6} > 0,$$

and so $\mathbb{L}_j(\mathbf{g}) \cap K \neq \emptyset$. In particular, for every $j \in [\![1, M]\!]$, there exists $x_j \in \mathbb{L}_j(\mathbf{g}) \cap K$ and so for all $i \in [\![1, M]\!]$

$$c(x_j, y_j) - g_j \leq c(x_j, y_i) - g_i \,.$$

Therefore, using the fact that the cost is non-negative, we have

$$\max_i g_i - \min_j g_j \leq \max_i \max_j \{c(x_j, y_i) - c(x_j, y_j)\}$$

$$\leq \max_{i \in [\![1, M]\!]} \sup_{x \in K} c(x, y_i) = \|c\|_{K, \infty} \,.$$

Moreover, since $H(\mathbf{g} + \lambda \mathbb{1}_M) = H(\mathbf{g})$ one can fix $g_1 = 0$, which concludes the proof. $\qquad \square$

# E Minimax estimation of OT quantities

Consider $P, Q \in \mathcal{P}(\mathbb{R}^d)$ with densities $f_P$ and $f_Q$. We recall that the Hellinger distance is defined by

$$d_{\mathrm{H}}(P, Q) := \left( \int_{\mathbb{R}^d} \left( \sqrt{f_P(x)} - \sqrt{f_Q(x)} \right)^2 \mathrm{d}\lambda_{\mathbb{R}^d}(x) \right)^{\frac{1}{2}}. \tag{17}$$

We also recall the formulation of Le Cam's Lemma. We refer to [35], Chapter 15, for more details.

**Lemma E.1.** *(Le Cam's Lemma.) Let $\mathcal{P}$ be a set of probability distributions on a measurable space, and consider the problem of estimating a parameter $\theta \in \Theta$ with a loss function $\ell$ defined, for all $\hat{\theta}, \theta \in \Theta$, as*

$$\ell(\hat{\theta}, \theta) = d(\hat{\theta}, \theta)^p,$$

*where $p \geq 1$ is an integer, and $d$ is a distance on $\Theta$. Then, for all $\theta_1, \theta_2 \in \Theta$,*

$$R_M = \inf_{T^{(n)}} \sup_{\theta \in \Theta} \mathbb{E}_\theta[\ell(\theta, T^{(n)}(X))] \geq \frac{1}{2^p} \left( 1 - \sqrt{n} d_{\mathrm{H}}(P_{\theta_1}, P_{\theta_2}) \right) d(\theta_1, \theta_2)^p,$$

*where $R_M$ is the minimax risk, and $T^{(n)}$ is an estimator based on $n$ i.i.d samples from $P_\theta$.*

In the sequel, we note $\mathcal{P}_{\mathrm{Lip}}$ the set of probability measures on $\mathbb{R}$ with Lipschitz densities.

## E.1 Kantorovich potential

*Proof.* Consider $\nu = \frac{1}{2}\delta_{\{0\}} + \frac{1}{2}\delta_{\{1\}}$ and fix the cost to transfer mass to be the usual quadratic cost $c(x, y) = \frac{1}{2}\|x - y\|^2$.

For $\delta \geq 0$, we define $\mu_{\theta\delta} = \mathcal{N}(\delta, 1)$. Since $d = 1$, the optimal transport map is monotone non-decreasing (see, for instance, Chapter 2 in [32]). Thus, we must have the identity

$$T_{\mu_{\theta_\delta}, \nu}(x) = \begin{cases} 0, & \text{if } x \leq \delta, \\ 1, & \text{otherwise.} \end{cases}$$

Therefore the vector $\theta^\delta \in \mathbb{R}^2$ solves the semi-dual problem if and only if it satisfies the following inequalities

$$c(x, 0) - \theta_1^\delta \leq c(x, 1) - \theta_2^\delta, \quad \forall x \leq \delta \,,$$
$$c(x, 1) - \theta_2^\delta \leq c(x, 0) - \theta_1^\delta, \quad \forall x > \delta \,.$$

Since $c(x, y) = \frac{1}{2}\|x - y\|^2$ we can fix $\theta_1^\delta = 0$ and compute

$$\theta^\delta = \left( 0, \frac{1}{2} - \delta \right).$$

Therefore, we parameterized the family of probability measures $\mu_\theta \in \mathcal{P}_{\mathrm{Lip}}$ so that the couple $(\theta^c, \theta)$ is the unique solution in $\Theta \subset \{\theta = (\theta_1, \theta_2) \in \mathbb{R}^2; \theta_1 = 0\}$ of the dual of $\mathrm{OT}(\mu_\theta, \nu)$. In this class of

probabilities, the minimax estimation of the optimal transport potential $\theta$, given $n > 0$ i.i.d samples of the source measure, can be written as

$$R_n^\Theta := \inf_{\hat{\theta}^{(n)}} \sup_{\theta \in \Theta} \mathbb{E}_{\mu_\theta}\left[\|\hat{\theta}^{(n)} - \theta\|^2\right],$$

where $\hat{\theta}^{(n)}$ is based on the $n$ i.i.d samples from the source measure $\mu$. Note that

$$R_n^\Theta \leq \inf_{\mathbf{g}^{(n)}} \sup_{\mu \in \mathcal{P}_{\mathrm{Lip}}} \mathbb{E}_\mu\left[\|\mathbf{g}^{(n)} - \mathbf{g}^*\|^2\right], \tag{18}$$

where the infimum is taken over all vectors $\mathbf{g}^{(n)}$ based on $n$ i.i.d samples of $\mu$.

Using the closed form of the Hellinger distance between Gaussian distributions, we have

$$d_{\mathrm{H}}(\mu_{\theta^0}, \mu_{\theta^\delta}) = \sqrt{1 - \exp\left(-\frac{\delta^2}{8}\right)}.$$

For $\delta \to 0$, the Taylor expansion gives $d_{\mathrm{H}}(\mu_{\theta^0}, \mu_{\theta^\delta}) = \delta/\sqrt{8} + o(\delta)$. Applying Le Cam's Lemma with $\delta = 1/\sqrt{n}$ gives

$$\begin{aligned}
R_n^\Theta &\geq \frac{1}{4}\left(1 - \sqrt{n}d_{\mathrm{H}}\left(\mu_{\theta^0}, \mu_{\theta^\delta}\right)\right)\|\theta^0 - \theta^\delta\|_2^2 \\
&\geq \frac{1}{4}\left(1 - 1/\sqrt{8} + o\left(1\right)\right)\frac{1}{n} \\
&\geq \frac{1}{10n} + o\left(\frac{1}{n}\right)
\end{aligned}$$

Using the inequality (18) concludes the proof. $\qquad\square$

### E.2  OT map

*Proof.* Define for $\delta \in [0, 1]$ the set of probability measures $\mu_\delta$, with density:

$$f_{\mu_\delta}(x) = \mathbf{1}_{x \in [0,1]}\left(1 + \delta g(x)\right), \qquad x \in [0, 1],$$

where

$$g(x) = \begin{cases} 2(1 - 2x), & x \in [0, 1/2], \\ -2(2x - 1), & x \in [1/2, 1]. \end{cases}$$

The squared Hellinger distance between $\mu_0$ (uniform) and $\mu_\delta$ is:

$$\begin{aligned}
d_{\mathrm{H}}(\mu_0, \mu_\delta)^2 &= \frac{1}{2}\int_0^1 \left(\sqrt{1} - \sqrt{1 + \delta g(x)}\right)^2 dx \\
&= \frac{1}{6\delta}\left((1 - 2\delta)^{3/2} + 6\delta - (1 + 2\delta)^{3/2}\right) \\
&= \frac{1}{2}\delta^2 + o(\delta^2)
\end{aligned}$$

when $\delta \to 0$. Therefore, we obtain

$$d_{\mathrm{H}}(\mu_0, \mu_\delta) = \frac{1}{\sqrt{2}}\delta + o(\delta).$$

Since $\mu_\delta \in \mathcal{P}_H$, $\delta \in [0, 1]$, we have

$$\inf_{T^{(n)}} \sup_{\mu \in \mathcal{P}_H} \mathbb{E}_\mu\left[\left\|T^{(n)} - T_{\mu,\nu}\right\|_{L^p(\mu)}^p\right] \geq \inf_{T^{(n)}} \sup_{\delta \in [0,1]} \mathbb{E}_{\mu_\delta}\left[\left\|T^{(n)} - T_{\mu_\delta,\nu}\right\|_{L^p(\mu_\delta)}^p\right]. \tag{19}$$

From the relation $1 - 2\delta \leq f_{\mu_\delta}(x) \leq 1 + 2\delta, \forall x \in [0, 1]$, we infer that no matter $T^{(n)}$ and $\delta$ small enough

$$\left\|T^{(n)} - T_{\mu_\delta,\nu}\right\|_{L^p(\mu_\delta)}^p \geq \frac{1}{2}\left\|T^{(n)} - T_{\mu_\delta,\nu}\right\|_{L^p(\mathcal{U}(0,1))}^p. \tag{20}$$

Consider $\nu = \frac{1}{2}\delta_{\{0\}} + \frac{1}{2}\delta_{\{1\}}$ as above. Recall that the optimal transport map is monotone non-decreasing in dimension 1. Moreover by definition $T_{\mu_\delta,\nu}(x) \in \{0,1\}$, $x \in [0,1]$, and $\mu_\delta(T_{\mu_\delta,\nu}^{-1}(0)) = 1/2$. Therefore one identifies $T_{\mu_\delta,\nu} = \mathbf{1}_{x \geq M_\delta}$ where $M_\delta$ is the median of $\mu_\delta$, $\delta > 0$, satisfying

$$\int_0^{M_\delta} \left(1 + \delta g(x)\right) dx = \frac{1}{2}.$$

Noticing that $M_\delta \in [0, 1/2]$, we solve

$$\int_0^{M_\delta} \left(1 + 2\delta(1 - 2x)\right) dx = \frac{1}{2},$$

which gives the solution

$$M_\delta = \frac{1 + 2\delta - \sqrt{1 + 4\delta^2}}{4\delta} = \frac{1}{2} - \frac{1}{2}\delta + o(\delta).$$

Observe that

$$\|T_{\mu_0,\nu} - T_{\mu_\delta,\nu}\|_{L^p(\mathcal{U}(0,1))}^p = \int_0^1 |\mathbf{1}_{x \geq M_0} - \mathbf{1}_{x \geq M_\delta}|^p \, dx = |M_\delta - M_0|.$$

We proved the relation $\|T_{\mu_0,\nu} - T_{\mu_\delta,\nu}\|_{L^p(\mathcal{U}(0,1))}^p = \delta + o(\delta)$. Applying Le Cam's Lemma with $\delta = \frac{1}{\sqrt{n}}$ for $n$ sufficiently large and any $p \in [1, \infty)$, we obtain

$$\inf_{T^{(n)}} \sup_{\delta \in [0,1]} \mathbb{E}_{\mu_\delta}\left[\left\|T^{(n)} - T_{\mu_\delta,\nu}\right\|_{L^p(\mathcal{U}(0,1))}^p\right] \geq \frac{1}{2^p}\left(1 - \sqrt{n}d_H(\mu_0, \mu_\delta)\right)\|T_{\mu_0,\nu} - T_{\mu_\delta,\nu}\|_{L^p(\mathcal{U}(0,1))}^p$$

$$\geq \frac{1}{2^p}\left(1 - \frac{1}{\sqrt{2}} + o(1)\right)\left(\frac{1}{\sqrt{n}} + o\left(\frac{1}{\sqrt{n}}\right)\right)$$

$$\gtrsim \frac{1}{\sqrt{n}}.$$

Therefore, combining inequalities (19) and (20) we conclude

$$\inf_{T^{(n)}} \sup_{\mu \in \mathcal{P}_H} \mathbb{E}_\mu\left[\left\|T^{(n)} - T_{\mu,\nu}\right\|_{L^p(\mu)}^p\right] \gtrsim \frac{1}{\sqrt{n}}.$$

$\square$

# F   Technical Lemmas

## F.1   Technical Lemmas for Appendix B

**Lemma F.1.** *Perturbation of Laplacian Matrices. Let $A$ and $B$ be symmetric Laplacian matrices of the same size such that:*

$$A_{ij} \leq 0, \quad B_{ij} \leq A_{ij} \quad \text{for all } i \neq j.$$

*Suppose $\lambda_2(A) > 0$, where $\lambda_2(A)$ denotes the second smallest eigenvalue of $A$. Then:*

$$\lambda_2(B) \geq \lambda_2(A)$$

*where $\lambda_2(B)$ is the second smallest eigenvalue of $B$.*

*Proof.* We recall the variational characterization of the second smallest eigenvalue of a Laplacian matrix $M$ is

$$\lambda_2(M) = \min_{x \perp \mathbf{1}} \frac{x^T M x}{x^T x}.$$

Define the matrix $C = B - A$. For $i \neq j$:

$$C_{ij} = B_{ij} - A_{ij} \leq 0 \,,$$

and for diagonal elements

$$C_{ii} = B_{ii} - A_{ii} = -\sum_{j \neq i} C_{ij} \geq 0 \,.$$

Thus, $C$ is a Laplacian matrix and so it is positive, semi-definite.

Let $y_2$ be the eigenvector corresponding to $\lambda_2(B)$, the second smallest eigenvalue of $B$, that is

$$B y_2 = \lambda_2(B) y_2 \quad \text{with } y_2 \perp \mathbf{1}.$$

Since $A = B - C$, we have

$$y_2^T A y_2 = y_2^T (B - C) y_2 = y_2^T B y_2 - y_2^T C y_2.$$

Thus,

$$\frac{y_2^T A y_2}{y_2^T y_2} = \lambda_2(B) - \frac{y_2^T C y_2}{y_2^T y_2}.$$

By the variational principle:

$$\lambda_2(A) = \min_{x \perp \mathbf{1}} \frac{x^T A x}{x^T x} \leq \frac{y_2^T A y_2}{y_2^T y_2} \leq \lambda_2(B) - \frac{y_2^T C y_2}{y_2^T y_2}.$$

Since $C$ is a Laplacian matrix, we have $y_2^T C y_2 \geq 0$, no matter $y_2$ and thus $\lambda_2(A) \leq \lambda_2(B)$ which completes the proof.

$\square$

**Lemma F.2.** *Let $f_\mu$ satisfy, for all $x \in \mathbb{R}^d$, the decay condition*

$$\sum_{r \geq 1} r^{d-1} \sup_{x \in \mathbb{R}^d \setminus B(0, r-1)} f_\mu(x) < \infty \,.$$

*Then, there exists a constant $C > 0$ such that, for any hyperplane $H \subset \mathbb{R}^d$, we have*

$$\int_H f_\mu(x) \, \mathrm{d}\mathcal{H}^{d-1}(x) \leq C.$$

*Proof.* Defining for any $r \in \mathbb{N}^*, R(r) := B(0, r) \setminus B(0, r - 1)$, we have

$$\int_H f_\mu(x) \mathrm{d}\mathcal{H}^{d-1}(x) = \sum_{r \geq 1} \int_{H \cap R(r)} f_\mu(x) \mathrm{d}\mathcal{H}^{d-1}(x)$$

$$\leq \sum_{r \geq 1} \int_{H \cap R(r)} \sup_{x \in R(r)} f_\mu(x) \mathrm{d}\mathcal{H}^{d-1}(x)$$

$$\leq \sum_{r \geq 1} \int_{H \cap R(r)} \sup_{x \in \mathbb{R}^d \setminus B(0, r-1)} f_\mu(x) \mathrm{d}\mathcal{H}^{d-1}(x)$$

Then, using the fact that $\mathcal{H}^{d-1}(H \cap R(r)) \leq \mathcal{H}^{d-1}(H \cap B(0, r))$, and noting that $H \cap B(0, r)$ is a $(d-1)$-dimensional ball of radius $r$, we have

$$\mathcal{H}^{d-1}(H \cap R(r)) \leq \frac{\pi^{(d-1)/2}}{\Gamma\left(\frac{d+1}{2}\right)} r^{d-1}.$$

Therefore, incorporating this bound, we obtain:

$$\int_H f_\mu(x) \mathrm{d}\mathcal{H}^{d-1}(x) \leq \sum_{r \geq 1} \frac{\pi^{(d-1)/2}}{\Gamma\left(\frac{d+1}{2}\right)} r^{d-1} \sup_{x \in \mathbb{R}^d \setminus B(0, r)} f_\mu(x)$$

which is finite by our decay assumption on $f_\mu$.

$\square$

### F.2 Technical Lemmas for Appendix C

**Lemma F.3.** *Let $(\gamma_n)_{n \geq 0}$ and $(m_n)_{n \geq 0}$ be some positive and decreasing sequences and let $(\delta_n)_{n \geq 0}$, satisfying the following:*

- *The sequence $\delta_n$ follows the recursive relation:*

$$\delta_{n+1} \leq (1 - \omega \gamma_{n+1}) \delta_n + m_{n+1} \gamma_{n+1}, \tag{21}$$

  *with $\delta_0 \geq 0$ and $\omega > 0$.*

- *$\gamma_n$ converges to $0$.*

- *Let $n_0 = \inf \{n \geq 1 : \omega \gamma_{n+1} \leq 1\}$, $\delta_{n_0}$ is non-negative.*

*Then, for all $n \geq n_0$, we have the upper bound:*

$$\delta_n \leq \exp\left(-\omega \sum_{i=n_0+1}^{n} \gamma_i\right)\left(\sum_{k=n_0}^{n} \gamma_k m_k + \delta_{n_0}\right) + \frac{1}{\omega} m_{\lceil \frac{n}{2} \rceil - 1}$$

*Proof.* For all $n \geq n_0$, one has

$$\delta_n \leq \underbrace{\prod_{i=n_0+1}^{n} (1 - \omega \gamma_i) \delta_{n_0}}_{=:U_{1,n}} + \underbrace{\sum_{k=n_0+1}^{n} \prod_{i=k+1}^{n} (1 - \omega \gamma_i) \gamma_k m_k}_{=:U_{2,n}}$$

One can consider two cases: $\lceil n/2 \rceil - 1 \leq n_0$ and $\lceil n/2 \rceil - 1 > n_0$.

**Case where $\lceil n/2 \rceil - 1 \leq n_0 < n$:** Since $m_k$ is decreasing,

$$U_{2,n} \leq m_{n_0+1} \sum_{k=n_0+1}^{n} \prod_{i=k+1}^{n} (1 - \omega \gamma_i) \gamma_k$$

$$= \frac{1}{\omega} m_{n_0+1} \sum_{k=n_0+1}^{n} \prod_{i=k+1}^{n} (1 - \omega \gamma_i) - \prod_{i=k}^{n} (1 - \omega \gamma_i)$$

$$= \frac{1}{\omega} m_{n_0+1} \left(1 - \prod_{i=n_0+1}^{n} (1 - \omega \gamma_i)\right)$$

$$\leq \frac{1}{\omega} m_{n_0+1}$$

Since $m_k$ is decreasing, it comes $U_{2,n} \leq \frac{1}{\omega} m_{\lceil n/2 \rceil}$.

**Case where $\lceil n/2 \rceil - 1 > n_0$:** As in [5], for all $m = n_0 + 1, \ldots, n$, one has

$$U_{2,n} \leq \exp\left(-\omega \sum_{k=m+1}^{n} \gamma_k\right) \sum_{k=n_0+1}^{m} \gamma_k m_k + \frac{1}{\omega} m_m$$

Then, taking $m = \lceil n/2 \rceil - 1$, it comes

$$U_{2,n} \leq \exp\left(-\omega \sum_{k=\lceil n/2 \rceil}^{n} \gamma_k\right) \sum_{k=n_0+1}^{\lceil n/2 \rceil - 1} \gamma_k m_k + \frac{1}{\omega} m_{\lceil n/2 \rceil - 1}$$

$\square$

**Lemma F.4.** *Linearization of the Hessian If a function $H : \mathbb{R}^M \to \mathbb{R}$ is such that its Hessian is $\alpha$-Hölder on the ball $B(0, r)$, with $r > 0, \alpha \in (0, 1)$ and constant $L$, then for any $\mathbf{g}, \mathbf{g}^* \in B(0, r)$, we have*

$$\|\nabla H(\mathbf{g}) - \nabla^2 H(\mathbf{g}^*)(\mathbf{g} - \mathbf{g}^*)\| \leq C \|\mathbf{g} - \mathbf{g}^*\|^{1+\alpha},$$

*where $C = \frac{L}{\alpha+1}$.*

*Proof.* Consider the Taylor expansion of $\nabla H(\mathbf{g})$ around $\mathbf{g}^*$:

$$\nabla H(\mathbf{g}) = \nabla H(\mathbf{g}^*) + \nabla^2 H(\mathbf{g}^*)(\mathbf{g} - \mathbf{g}^*) + R(\mathbf{g}),$$

where the remainder term $R(\mathbf{g})$ is given by:

$$R(\mathbf{g}) = \int_0^1 \left[\nabla^2 H(\mathbf{g}^* + t(\mathbf{g} - \mathbf{g}^*)) - \nabla^2 H(\mathbf{g}^*)\right](\mathbf{g} - \mathbf{g}^*)\mathrm{d}t.$$

By the assumption of $\alpha$-Hölder continuity of the Hessian, we have

$$\|\nabla^2 H(\mathbf{g}^* + t(\mathbf{g} - \mathbf{g}^*)) - \nabla^2 H(\mathbf{g}^*)\| \le L\|t(\mathbf{g} - \mathbf{g}^*)\|^\alpha = Lt^\alpha\|\mathbf{g} - \mathbf{g}^*\|^\alpha.$$

Thus,

$$\|R(\mathbf{g})\| \le \int_0^1 Lt^\alpha\|\mathbf{g} - \mathbf{g}^*\|^\alpha\|\mathbf{g} - \mathbf{g}^*\|\mathrm{d}t = L\|\mathbf{g} - \mathbf{g}^*\|^{1+\alpha}\int_0^1 t^\alpha\mathrm{d}t.$$

Evaluating the integral:

$$\int_0^1 t^\alpha\mathrm{d}t = \frac{1}{\alpha + 1}.$$

Therefore,

$$\|R(\mathbf{g})\| \le \frac{L}{\alpha + 1}\|(\mathbf{g} - (\mathbf{g}^*\|^{1+\alpha}.$$

This implies the desired inequality:

$$\|\nabla H(\mathbf{g}) - \nabla^2 H(\mathbf{g}^*)(\mathbf{g} - (\mathbf{g}^*)\| \le C\|(\mathbf{g} - (\mathbf{g}^*\|^{1+\alpha},$$

where $C = \frac{L}{\alpha+1}$. $\qquad\square$

