# OpenReview forum: "Stochastic Optimization in Semi-Discrete Optimal Transport: Convergence Analysis and Minimax Rate"
_NeurIPS.cc/2025/Conference — NeurIPS 2025 spotlight_

### Official Review · Reviewer_M6v8 · 2025-06-11

**Clarity:** 3
**Significance:** 2
**Originality:** 3
**Rating:** 5
**Confidence:** 2

**Summary:**

Semi-discrete optimal transport refers to the scenario where we would like to learn the optimal transport map that transports a continuous source measure into a discrete target measure. This problem can be tackled using Stochastic Gradient Descent (SGD), which results in empirical successes but it is unclear how it is able to work so well. The paper considers the projected version of SGD (PSGD), and provide convergence guarantees of PSGD. There are convergence guarantee of PSGD in the general scenario, convergence guarantee of PSGD for MTW cost and compact case and for quadratic cost and non-compact case, among others. It was demonstrated that PSGD can learn the optimal transport map reasonably efficiently. Numerical experiments are provided, verifying the theoretical claims.

**Questions:**

Please see the weakness in Strengths and Weaknesses section.

Some minor issues:

1. In TL;DR on OpenReview, "in the semi-discre setting" -> "in the semi-discrete setting"
2. MTW properties was mentioned on Line 82, while introduced only on Line 93
3. Line 146: "[XXXX]" bibtex error perhaps?
4. Theorem 5.3, the equation should end with "," instead of "."

**Ethical Concerns:**

["NO or VERY MINOR ethics concerns only"]

**Final Justification:**

I found the theory as outlined in the paper interesting and could not found significant issues apart from the error bars missing. As the authors have promised to add the error bars, I maintain my original positive rating.

**Limitations:**

yes

**Paper Formatting Concerns:**

The checklist guidelines were deleted while the instructions state that they should be kept.

**Quality:**

3

**Strengths And Weaknesses:**

Strengths:

1. The research questions are well-motivated.
2. The theoretical results are rigorous, demonstrating the reasonable convergence of PSGD for semi-discrete OT.

Weaknesses:

1. For the experiments, no error bars were provided: what are the standard deviations across the runs?

---

> ### Author Rebuttal · Authors · 2025-07-30
>
> We sincerely thank the reviewer for their thoughtful and positive assessment of our work. We appreciate the recognition of the motivation and theoretical contributions in our paper.
>
> **Error bars in experiments.**
> Thank you for pointing this out. We did look at the standard deviations (based on multiple runs of the SGD-based algorithms), and they are consistently small, especially for the averaged iterates and the OT map estimation, indicating low variance in the performance. We agree that showing this is important and will include error bars in the revised version to make this clearer to the reader.
>
> **Minor issues.**
> We will correct the typo in the TL;DR , fix the BibTeX citation issue at line 146, and update the punctuation in Theorem 5.3. We will also make sure that the introduction of the MTW properties comes before they are first mentioned (currently line 82), to improve clarity. Thank you for catching these details.
>
> **Formatting.**
> We will restore the checklist guidelines as required.

---

> > ### Comment · Reviewer_M6v8 · 2025-08-03
> >
> > I thank the authors for their response. I would like to maintain my original positive rating.

---

### Official Review · Reviewer_9oFK · 2025-07-01

**Clarity:** 2
**Significance:** 3
**Originality:** 3
**Rating:** 4
**Confidence:** 3

**Summary:**

This paper studies the semi-discrete optimal transport problem, whereby one seeks to compute the optimal transport cost and map between a continuous distribution and a discrete one. The authors attack the problem using stochastic gradient descent and study their computational and statistical properties, establishing the convergence of optimal transport costs, Kantorovich potentials, and optimal transport maps with a rate of $1/\sqrt N$ even when the continuous measure does not have compact support. In a specific setting, the authors also show that the rate for optimal transport maps is minimax optimal. Numerical experiments support the theoretical findings.

**Questions:**

1. Can the authors please clarify the position of their paper w.r.t. the literature (see comment above)?
2. Can the authors please clarify my concerns for Lemma 3.1 and Theorem 5.3 (see comment above)?
3. Can the authors please clarify assumption B (see comment above)?
4. The authors repeatedly used “linear convergence”. What do they mean by that? The algorithm does not enjoy “linear convergence” as commonly meant in the optimization literature.
5. In Section 6  and Fig. 3c, the empirical results do not seem to precisely match the theoretical bound, at least not as accurately as in the other experiments. Can the authors explain?

**Ethical Concerns:**

["NO or VERY MINOR ethics concerns only"]

**Final Justification:**

The authors have satisfactorily addressed my concerns (in particular, Theorem 5.3 and positioning w.r.t. the literature). Thus, provided that these comments are incorporated in the revision, I am happy to increase my score.

**Limitations:**

Yes.

**Paper Formatting Concerns:**

None.

**Quality:**

2

**Strengths And Weaknesses:**

**Strenghts:**
- The paper studies an important problem, namely the design of computational algorithms for the semi-discrete optimal transport, which is relevant for a variety of machine learning applications.
- The algorithm proposed by the authors is intuitive and simple to implement, and enjoys rigorous theoretical guarantees of convergence (rate of $1/N$ for the squared error of the potentials and $1/\sqrt N$ for OT maps, under mild assumptions).
- The paper includes several technical results, in particular boundedness of Kantorovich potentials in the non-compact case (Lemma 3.1) and Lipschitzness of the “error” in the optimal transport map in the non-compact case (Theorem 5.1) which are of independent interest.

**Weaknesses and comments:**
-  The paper should significantly improve its positioning w.r.t. the existing literature, especially in the following two ways:
  - What are the novel contributions of the paper compared to [5, 20, 29]? The paper definitely contains novel contributions (e.g., the non-compact case), but the existing work should be better clarified (not only with a citation but with references to the statements). Also, it seems that [29] considers the entropic optimal transport problem.
  - The authors should review existing results on the regularity of Kantorovich potential and their boundedness.
  - There is existing work by Taşkesen and co-authors (see reference below and references therein) on the computational aspects of semi-discrete optimal transport. The authors should position their work w.r.t. this literature.
- Lemma 3.1 and its proof:
  - I suspect Lemma 3.1 cannot be true without assumptions on transportation cost; indeed, if the transportation cost fails to be lower semi-continuous, then Kantorovich duality might fail (indeed, reference [31] assumes lower semi-continuity of the transportation cost to derive duality). Thus, I suppose some regularity of the transportation cost is necessary.
  - The authors say “suppose that (4) admits a minimum”, but already write max in (3). In fact, Kantorovich potentials are well-defined under minor assumptions on the transportation cost, so I do not see the benefit of being more general here.
  - The equation after line 954 in the proof of Lemma 3.1 needs to be better explained, saying that it follows from the definition of $\mathbb{L}$.
- The results in the non-compact case are predicated under Assumption B, which is however never discussed in the paper (in particular, (B3), which does not have an analog in the compact case). Given that the focus that the authors put the non-compact case, the assumption must be explained and justified.
- Theorem 5.3:
  - I fail to understand why Theorem 5.3 is proven in the one-dimensional case when $\nu$ is only supported on two points.
  - I fail to understand why the proof implies the presented result. For instance, the set $\mathcal{P}_\mathrm{lip}$ appears in the statement of the theorem, but never in the proof.
  - In the proof, the authors focus on Gaussian distributions (in E1) and specific distributions (in E2). I do not see why this continues to imply the general result in Theorem 5.3.
- Minor comments:
  - The structure of the appendix of the paper is confusing, with some of the results hard to find (e.g., the proof of Theorem 5.3 is not clearly identified, same for the one of proposition 4.1 and Lemma 4.2).
  - Defining measures as on line 73 is not rigorous. A measure “eats” sets of reals and not reals.
  - The acronym MTW is used in the abstract and section 2 before being defined.
  - There is a [XXXX] on line 146.
  - In Section 6, $\mathbf{g}$ should be $\mathbf{g}^\ast$, in lines 317-318.
  - The norms and inner product defined in the equation after line 219 are in fact seminorms. I suggest making this point clear in the text and in the proofs (e.g., Proposition C1).
  - In Proposition B1, the term $\mathcal{H}$ appears undefined.
  - In Proposition C1, there is a 2 missing in the exponent in the left-hand side of the equation (but the proof appears correct).
  - In the appendix, Theorem 4.5 is in fact 4.4 and Theorem 4.4 is 4.5.
  - In the proof of Theorem 4.5., $p_k, \xi_{k+1}, \delta_k$ are vectors. I suggest to use bold to unify the notation.
  - $\bar g_n$ should be bold in the equation after line 934.
  - In the equation after line 955, I suspect that the last sum should run from $k=1$ to $n+1$ (and not $n$), implying that the last term is     $8\gamma_1\sqrt{n+2}$.
  - There should no max over j in the second line of equation 955.
  - Many equations in the appendix go out of margin, and one equation (after line 939) does not remain visible.
- Typos:
  - Period missing on line 47.
  - There should be no period at end of line 607 and a period at the end of line 613.


Taşkesen, Bahar, Soroosh Shafieezadeh-Abadeh, and Daniel Kuhn. "Semi-discrete optimal transport: Hardness, regularization and numerical solution." Mathematical Programming 199.1 (2023): 1033-1106.

---

> ### Author Rebuttal · Authors · 2025-07-30
>
> We sincerely thank the reviewer for their thoughtful, detailed, and constructive feedback. Your comments have been extremely helpful and have led us to significantly improve the clarity, rigor, and positioning of our paper. Your suggestions helped us identify several key areas requiring clarification, and we are confident that the revised version of the paper will be much stronger as a result.
>
> We respond point by point below.
>
>
> ### **Questions**
>
>
> **1.** Positioning w.r.t. the existing literature:
> We understand the reviewer's concerns and will include a more thorough comparison with the literature, notably by highlighting the following points.
>
> * **Generalizing properties of semi-discrete OT to the non-compact case:** In [5, 20], the authors present several properties of the semi-dual and
> semi-discrete OT, which we recall in Appendix B.1. In our work, we notably generalize these properties to the non-compact case for the quadratic cost.
>  * **Giving the first minimax results for non-quadratic cost and non-compact case in
> semi-discrete OT, and moreover with an online/one-pass algorithm:**
> In [29], the authors consider the estimation of the OT map from samples, but only
> in the compact case and for the quadratic cost. To achieve this, they show that the
> discrete entropic map, with regularization $\varepsilon \simeq 1/\sqrt{n}$,
> approximates the true OT map with an error of $\mathcal{O}(1/\sqrt{n})$.
> This entropic discrete map can be approximated arbitrarily closely using,
> for instance, the Sinkhorn algorithm.
> In our paper, we address OT map estimation for a broader class of cost functions,
> including non-quadratic costs and the non-compact setting. However, we assume
> full access to the discrete measure (one-sample setting), whereas their estimator
> achieves the same rate even when sub-sampling from $\nu$ (two-sample setting).
> A key strength of our estimator is its ability to operate in an online setting,
> without the need to store samples drawn from $\mu$, and to refine the estimate
> as more samples become available.
> While some of this information appears in lines 303–309, we will emphasize
> these differences more clearly in the revised version.
> * **Studying SGD algorithms for non regularized semi-discrete OT:**
> In Taşkesen et al., the authors study regularized optimal transport in the
> semi-discrete setting, with a class of regularizers that notably includes
> entropic regularization. They provide convergence guarantees for the regularized
> objective function.
> In contrast, we focus on the non-regularized semi-discrete OT problem, and
> analyze the convergence of both the unregularized OT map and cost. We will
> incorporate a discussion of this work into the paragraph currently titled
> "Entropic regularization" (starting at line 124), which we will rename
> "Regularization of the semi-dual".
>
>
>
> **2.**  Lemma 3.1:
> You are right, duality may fail without assumptions on the regularity of the cost. However, the existence of a minimizer in the stated set is still guaranteed, provided that (4) admits a minimizer. That said, we agree that minimizing (4) is not meaningful if there is a duality gap. We will therefore restate the theorem under the assumption that (3) is well-posed and that strong duality holds, as is the case when the cost is lower semi-continuous and bounded from below.
>
> We will also add details after line 954, showing that $H$ is invariant under constant shifts of the potential.
>
> **3.** **A)** Assumption B (non-compact case)
> We agree that Assumption B3 is new and somewhat technical. However, it is satisfied by common distributions such as the Normal distribution and the Student distribution with degrees of freedom $\nu > 2$.
> We will explain that it is necessary for the definition and regularity of the Hessian of the semi-dual.
> We will add an example in the appendix showing that if B3 fails, the Hessian may not be defined. This example uses a measure with a spiky density of the form
>
> \begin{equation*}
> f_\mu(x) = \frac{1}{Z} \left[ \frac{1}{1 + \|x\|^4} + \sum_{k=1}^\infty k \cdot \mathbf{1}_{R_k}(x) \right]
> \end{equation*}
> where $R\_{k} := \{ (x\_1, x\_2) :  |x\_1| \le \tfrac{1}{k^5},\; k \le x\_2 \le k+1 \}$
> and  $Z$ is the normalizing constant.
> A complete proof of this counter-example will be given in the appendix, and we will refer to it when introducing Assumption B.
>
> **B)** (Theorem 5.3)
> * This theorem states that even in the one-dimensional case, where the source measure
> $\nu$ is as simple as $\nu =\frac{1}{2}\delta_{\{0\}} + \frac{1}{2}\delta_{\{1\}}$,
> no estimator based on $n$ i.i.d. samples can achieve a convergence rate better than
> $\mathcal{O}(1/\sqrt{n})$ uniformly over all Lipschitz measures.
> Fixing the measure $\nu$ while considering a class of source measures is analogous
> to the minimax setting in [29, Proposition 1], where the roles are reversed: the source
> measure is fixed and a class of discrete target measures is considered.
> *  There is a typo in Appendix E concerning the proofs of the minimax lower bounds:
> we consistently refer to $\mathcal{P}\_{H}$ instead of $\mathcal{P}_{\mathrm{Lip}}$.
> We will correct this typo in the revised version and apologize for the inconvenience.
> We focused on Lipschitz measures to emphasize that even within this more regular class,
> the minimax rate remains the same. Since the class of measures with Lipschitz density
> is a subclass of those with $\alpha$-Hölder density for $\alpha \in (0,1]$,
> which we study in the paper, the lower bound extends to this broader class as well.
> We will clarify this point more explicitly in the revised version.
> * The proof uses Le Cam’s two-point method. The idea is that to lower-bound the minimax risk,
> it suffices to construct a pair of distributions in the model class that are both hard to
> distinguish (in Hellinger distance) and induce notably different values of the quantity
> to be estimated. The lower bound obtained for this pair then extends to the entire class
> they belong to.
> In Appendix E.2, which contains the proof of Theorem 5.3, we consider a subclass of Lipschitz
> probability measures $\mu_\delta$, $\delta \geq 0$, and our analysis through (19) and (20)
> yields the lower bound
>
> $$
> \inf_{T^{(n)}} \sup_{\mu \in \mathcal{P}_{\mathrm{Lip}}} \mathbb{E}\_{\mu}[||T^{(n)} - T\_{\mu, \nu}||\_{L^p(\mu)}^p] \geq \frac{1}{2} \inf\_{T^{(n)}} \sup\_{\delta \in [0,1]}\mathbb{E}\_{\mu\_{\delta}}\big[||T^{(n)} - T\_{\mu\_{\delta}, \nu}||\_{L^p(\mathcal{U}(0,1))}^p\big]
> $$
>
> The goal of this inequality is to eliminate the dependence on $\mu$ in the norm
> $||\cdot||\_{L^{p}(\mu)}$ by replacing it with the fixed reference measure $\mathcal{U}(0,1)$
> on the right-hand side. This allows us to treat $||\cdot||\_{L^p(\mathcal{U}(0,1))}$
> as the loss function $l$ in the statement of Le Cam's Lemma (Lemma E.1).
> We then apply Le Cam’s two-point method to the pair $(\mu_0, \mu_\delta)$ and propagate
> the resulting lower bound to the full class. We will ensure that the proof is made
> more explicit and reader-friendly in the revised version.
>
> **4.** Linear rate:
> We apologize for any confusion. We referred to $\mathcal{O}(1/t)$ as a linear rate, which differs from the standard use of the term "linear" in the deterministic optimization literature, where it typically denotes exponential convergence. In the revised version, we will avoid using the term "linear rate" to describe $\mathcal{O}(1/t)$ convergence.
>
>
> **5.**  In Section 6, Fig. 3c, we believe that the small discrepancy comes from the fact that the
> errors $||\bar{\mathbf{g}}\_t - \mathbf{g}^{\*}||^2$ are already below $10^{-4}$.
> The target weights of $\nu$ (such that the randomly generated $\mathbf{g}^*$ is optimal)
> are approximated using a Monte Carlo estimator with $10^9$ samples. At that scale,
> the residual error in approximating the optimal weights contributes to the slightly less
> sharp convergence behavior observed in this plot. Indeed, the Monte Carlo error for each
> weight is expected to be on the order of $\mathcal{O}(1/\sqrt{N})$ with $N$ samples,
> and this propagates into the final error metric. To validate this hypothesis, we reran
> the experiments using $10^{11}$ samples for the Monte Carlo approximation. With this
> higher precision, the convergence plot aligned closely with the others, confirming our
> interpretation. We will update the figure to include the higher-precision results.
>
>
>
> ### **Minor comments**
>
> We agree with the minor comments and thank the reviewer for the detailed suggestions. We will correct the typos, notably by defining $\mathcal{H}^{d-1}$ as the $(d{-}1)$-dimensional Hausdorff measure in Proposition B.1, clarifying the use of seminorms, unifying the vector notation, and ensuring that all symbols and acronyms are properly defined.
> We will also reorganize the appendix for clarity, fix formatting issues, ensure that each theorem has a dedicated proof with an explicit section title, and adjust incorrect indices or equations as pointed out.

---

> > ### Comment · Reviewer_9oFK · 2025-08-04
> >
> > I thank the authors for their detailed response. All my questions are answered satisfactorily, and I am happy to increase my score.

---

### Official Review · Reviewer_74ib · 2025-07-01

**Clarity:** 3
**Significance:** 3
**Originality:** 2
**Rating:** 5
**Confidence:** 3

**Summary:**

Summary: This paper considers the semi-discrete optimal transport problem where we are given a discrete distribution \mu and a continuous distribution \nu and one wishes to compute the optimal transport from \mu to \nu. The semi-discrete optimal transport is more expressive than the discrete OT and more tractable than the continuous OT. Moreover, unlike the continuous OT, estimating the semi-discrete OT via samples does not suffer from the curse of dimensionality. In fact, with n samples, we can get an estimate of the semi-discrete OT cost and map with a convergence rate of 1/\sqrt{n} (I believe this convergence rate hides the support size of the discrete distribution).

From an algorithmic stand-point, even with the sampling result, one has to compute the exact discrete OT to estimate the cost. The authors suggest SGD based algorithm that uses these samples and estimates the cost and plan efficiently. The authors also complement their theoretical results with an implementation and analyze the performance.
Another major contribution of this paper is in reducing the assumptions (for instance, compactness assumption) from the convergence rate.

The authors, however, miss an important reference which provides an alternate approach for low-dimensional settings (the result presented at the end of [1]). In this alternate approach, one can pre-process the sample in a data structure in time that is linear in the number of samples, and obtain the exact OT cost (between discrete distributions) using only sub-linearly many calls to this data structure. So, by combining this with the result of Pooladian et al., one should be able to get a linear-time estimator in low dimensions.

Overall, I think this is a solid paper that can be accepted.

[1] 	Pankaj K. Agarwal, Sharath Raghvendra, Pouyan Shirzadian, Keegan Yao:
A Combinatorial Algorithm for the Semi-Discrete Optimal Transport Problem. NeurIPS 2024

**Questions:**

Please answer the questions I raised in weakness.

**Ethical Concerns:**

["NO or VERY MINOR ethics concerns only"]

**Final Justification:**

This paper studies the behavior of an important approach (SGD) for an important problem (semi-discrete OT) and provides some useful bounds. The authors answered my questions satisfactorily and I maintain my positive score for the paper.

**Limitations:**

Yes

**Paper Formatting Concerns:**

No major concerns

**Quality:**

3

**Strengths And Weaknesses:**

Positives: This paper establishes near optimal convergence rate for SGD based approaches to estimate both the semi-discrete OT cost and map. The paper generalizes the convergence rate to various cost functions as well as for distributions that are not necessarily compact.

One the flip side, the paper does not discuss the dependence of the convergence rate on the size of the discrete support. Could the authors provide the dependence of the convergence rate as a function of the size of the support of discrete distribution?

There are also a few typos across the paper. For instance, L13 (whic -> which) and L146 ([XXXX] must be removed)

---

> ### Author Rebuttal · Authors · 2025-07-30
>
> We sincerely thank the reviewer for their constructive and positive feedback. Thank you for pointing out typos, we will correct them for the final version. We will address the specific questions and comments raised below.
>
>
> 1. Thank you for pointing out the algorithm described at the end of [1]. It is true that their method achieves linear computational complexity in the number of samples and could serve as an alternative to the Sinkhorn algorithm to obtain an efficient estimator in dimension 2, in the setting of Pooladian et al., i.e., for the quadratic cost with a compactly supported source measure. While the statistical/sample complexity remains the same as in Pooladian et al., their algorithm could offer faster runtime than Sinkhorn. We will make sure to cite this work in the revised version.
>
> 2. In our convergence analysis (Appendix C.2, see the equation after line 925), we show:
>
> $$
> \mathbb{E}\left[\| \bar{\mathbf{g}}_n - \mathbf{g}^* \|_v^{2} \right] = \frac{1}{\lambda^2(n+1)} + o\left( \frac{1}{n} \right),
> $$
>
> where $\lambda > 0$ is the smallest eigenvalue of the Hessian of $H$ at the optimum, characterizing the local strong convexity of $H$.
>
> This was briefly noted between lines 271 and 274. While prior work has analyzed $\lambda$ in specific cases, it is known to depend on the structure of both $\mu$ and $\nu$. However, we do not currently have general estimates for its value, and obtaining such bounds remains a challenging and interesting direction for future research.
>
> We agree that this aspect deserves more attention. After Theorem 4.5, we will clarify that although the convergence rate depends on $\lambda$, sharp bounds on its dependence (e.g., in terms of the number of points) are not yet available in full generality.
>
> Nonetheless, combining this dependence with our stability results on Laguerre cells, we can derive,for instance, when $f\_\mu$ is $\alpha$-Hölder as in Corollary 2.2, a convergence of the OT map of $\mathcal{O}\left( \frac{M^2 }{\lambda\sqrt{n}} \right)$.
>
> Although the dependence on $M$ that comes from our analysis is conservative, we will include and highlight this bound in the revised version.

---

> > ### Comment · Reviewer_74ib · 2025-08-04
> > **Thank you**
> >
> > Thank you for addressing my questions. I recommend that the introduction include a discussion of how the convergence rate and execution time of your algorithm depend on the size of the discrete distribution's support and the problem's dimension, as well as what happens when your assumptions are not satisfied. Additionally, please include a comparison with the work of Agarwal et al., NeurIPS 2024, which operates in fixed dimensions without making assumptions on the continuous distribution.
> >
> > I believe it is plausible that, under reasonable assumptions on the continuous distribution, the algorithm by Agarwal et al., NeurIPS 2024 may exhibit significantly better dependence on the dimension.
> >
> > I remain positive about this submission and maintain my favorable score.

---

> > > ### Author Response · Authors · 2025-08-08
> > > **Thank you for your reply**
> > >
> > > Dear Reviewer,
> > > Thank you again for your thoughtful follow-up and for maintaining a positive assessment of our submission.
> > >
> > > We appreciate your suggestion to clarify how the convergence rate and runtime of our algorithm depend on the support size and dimension, as well as the effects of violating our assumptions. We will incorporate this discussion into the introduction to better contextualize the scope of our contributions.
> > >
> > > Regarding the algorithm by Agarwal et al. (NeurIPS 2024), we agree that their method is an interesting contribution, especially as it makes no assumptions on the smoothness of the continuous distribution. That said, their Theorem 1.2 achieves an $\varepsilon$-accurate OT plan for the quadratic cost \$c(x,y) = \frac{1}{2}||x - y||^2\$ with a convergence rate of
> > > $$
> > > O\left(M^{d+1} N^{1 - 1/d} \log \frac{1}{\varepsilon} \right),
> > > $$
> > >
> > > where \$M\$ and \$N\$ denote the number of points in the target and (discretized) source measures, respectively. This implies an exponential dependence on the dimension.
> > >
> > > In contrast, our algorithm operates in an online setting without requiring discretization or storage of past samples, and achieves \$\varepsilon\$-accuracy in
> > >
> > > $$
> > > \mathcal{O}\left( \frac{M^2}{\lambda \varepsilon} \right)
> > > $$
> > >
> > > iterations. While the regularity constant \$\lambda\$ may vary, existing theoretical estimates (as referenced in our paper) and empirical performance, including in dimension 100, suggest that \$\lambda\$ does not scale exponentially with \$d\$.
> > >
> > > Overall, we believe this reflects a complementary trade-off: their method has polynomial dependence on \$\log(1/\varepsilon)\$ but exponential in dimension, whereas ours is exponential in \$\log(1/\varepsilon)\$ but polynomial in dimension. This distinction highlights that their method may be better suited for low-dimensional problems, while ours remains effective in higher dimensions and streaming contexts. We also note that achieving polynomial dependence in both \$\log(1/\varepsilon)\$ and dimension is known to be NP-hard \[1], which supports the plausibility of this trade-off.
> > >
> > > Thank you again for your insightful comments and for engaging deeply with our work.
> > >
> > > **Reference:**
> > >
> > > [1]Taşkesen, Bahar, Soroosh Shafieezadeh-Abadeh, and Daniel Kuhn. "Semi-discrete optimal transport: Hardness, regularization and numerical solution." Mathematical Programming 199, no. 1 (2023): 1033-1106.

---

> ### Comment · Reviewer_74ib · 2025-08-08
>
> My earlier comment lacked sufficient context. Here is what I intended:
>
> In fully discrete settings, the combinatorial graph techniques of Gabow and Tarjan, (SICOMP 1989), yield an $n^{2.5} \log (1/\varepsilon)$-time algorithm. Lahn et al., (NeurIPS 2019) demonstrated that a mild modification of these techniques leads to a different trade-off between $n$ and $\varepsilon$, achieving a running time of $n^2 / \varepsilon$.
>
> Analogously, although it would be non-trivial, the combinatorial framework introduced by Agarwal et al., NeurIPS 2024 might be adaptable to obtain a different trade-off between $d$ and $\varepsilon$, particularly under reasonable assumptions on the continuous distribution.
>
> SGD-based algorithms are also of significant value, and understanding their behavior is an important contribution that this paper advances.

---

> > ### Author Response · Authors · 2025-08-08
> > **Thank you for the additional context**
> >
> > We are grateful for the reviewer’s follow-up and helpful clarification. We will ensure that the reference to Agarwal et al., NeurIPS 2024 is appropriately emphasized in the introduction of the revised manuscript.

---

### Official Review · Reviewer_g7BH · 2025-07-03

**Clarity:** 2
**Significance:** 4
**Originality:** 3
**Rating:** 5
**Confidence:** 3

**Summary:**

The paper studies the convergence of an online SGD-type method for the semi-discrete optimal transport problem, where the source measure $\mu$ is discrete and the target measure $\nu$ is continuous. The analysis is applicable to a broad range of settings, including: 1) cases where costs satisfy the Ma-Trudinger-Wang properties (MTW), and $\operatorname{supp}\mu$ is compact; 2) cases where costs are quadratic (in which case compactness of $\operatorname{supp}\mu$ is not required). The authors consider a semi-dual formulation and introduce a compact set $\mathcal{C}$, which they prove contains a solution. This set is then used as a projection set in the iterations of an SGD-type method. This leads to an $\mathcal{O}(1/\sqrt{n})$ rate for OT map estimation and $\mathcal{O}(1/n)$ rates in terms of the objective and potential. The authors prove that these rates are minimax optimal. Numerical experiments illustrate the performance of the algorithm.

**Questions:**

1. In which cases is the compact set $K$ (used in defining $\mathcal{C}$) difficult or impossible to identify?
2. The objective $H$ is referenced in lines 95-96 prior to its formal definition, exemplifying clarity issues in the presentation.
3. Please correct typos (e.g., replace "whic" with "which" in line 13).

The paper's final score will depend on satisfactory resolution of these issues.

**Ethical Concerns:**

["NO or VERY MINOR ethics concerns only"]

**Final Justification:**

Apart from minor remarks, I only had one question which the authors addressed in detail. The paper's contribution seems valuable and I don't see any major issues. Therefore, I maintain my positive score.

**Limitations:**

The limitations related to finding the set $K$ may be discussed in more detail. Apart from that, limitations are mostly addressed.

**Paper Formatting Concerns:**

No major formatting issues

**Quality:**

3

**Strengths And Weaknesses:**

Strengths are as follows:
- The theoretical analysis of the method is applicable to a broad range of settings,
- Lower bounds are established,
- Numerical experiments illustrate the performance of the method.

As for the weaknesses, the clarity and quality of the presentation should be improved, and typos should be corrected. In my opinion, more discussion about the compact set $K$ used in the definition of $\mathcal{C}$ is needed (see questions).

---

> ### Author Rebuttal · Authors · 2025-07-30
>
> We thank the reviewer for the thoughtful and positive feedback. We respond below to the specific concerns raised.
>
> **Question 1:** While the compact set $K$ is easy to find analytically when the density is known, we can always identify, with high probability, a ball $B(0,R)$ such that $\mu(B(0,R)) \geq 1 - \frac{1}{4}w_{\min}$, and therefore use the compact set $K = B(0,R)$, even when only samples from $\mu$ are available without any other information.
> Indeed, this collapses to a classical problem of estimating a one-dimensional quantile.
> Write $X\sim\mu$ and define
>
> $$
> R\_{\star} = \inf\big\\{r:\; \mu(||X|| \le r) \ge 1 - \tfrac{1}{4} w\_{\min} \big\\}
> = F^{-1}\big(1 - \tfrac{1}{4} w\_{\min} \big)
> $$
>
> where $F(r)=\mu(||X||\le r)$ is the cumulative distribution function of the scalar random variable $||X||$.
> Because we do not need the exact infimum, and any $R \geq R_\star$ provides a satisfactory $K = B(0,R)$, we estimate a slightly higher quantile.
>
> Define the estimator
>
> $$
> \widehat R\_{\eta} := \inf\\{\,r:\; F\_n(r) \ge 1 - \tfrac{1}{4} w\_{\min} + \eta\\},
> \qquad 0 < \eta < \tfrac{1}{4} w\_{\min}
> $$
>
> where $F_n(r)=\frac1n\sum_{i=1}^n \mathbf{1}_{\{||X_i||\le r\}}$ is the empirical CDF of $||X||$.
>
> By the Dvoretzky–Kiefer–Wolfowitz inequality with Massart’s sharp constant [1,2],
> choosing  $ \eta = \frac{1}{8}w_{\min},
> n\ge\frac{\log(2/\delta)}{2\eta^{2}}
> $
> ensures that, with probability at least $1-\delta$, we have $
> \mu\bigl(B(0,\widehat R_{\eta})\bigr)\ \ge\ 1-\tfrac14 w_{\min}
> $,
> which is a dimension‑free sample‑complexity bound.
>
> We note that tighter bounds in terms of $\eta$ may be achievable, but we rely here on well-established classical results for clarity and generality.
>
> We thank the reviewer for this important question. Due to space constraints, we will include the full argument and dimension-free sample complexity bound in the supplementary material, and add a concise summary in the main paper to clarify that such a compact set $K$ can always be estimated from samples with high probability.
>
> **Question 2:** You are correct, the reference to $H$ appears to early and may cause confusion. We will revise the text to ensure $H$ is properly introduced before being used, in order to improve clarity.
>
> **Question 3** Thank you for pointing out the typos. We have corrected them in the revised version.
>
> ### **References**
> [1] Dvoretzky, A., Kiefer, J., \& Wolfowitz, J. (1956). Asymptotic minimax character of the sample distribution function and of the classical multinomial estimator. Annals of Mathematical Statistics, 27(3), 642–669.
>
> [2] Massart, P. (1990). The tight constant in the Dvoretzky–Kiefer–Wolfowitz inequality. Annals of Probability, 18(3), 1269–1283.

---

> > ### Comment · Reviewer_g7BH · 2025-08-08
> >
> > I thank the authors for addressing my questions and maintain the positive score

---

### Decision · Program_Chairs · 2025-09-17

**Decision:**

Accept (spotlight)

**Comment:**

The paper addresses the semi-discrete OT (continuous source transported to discrete target), showing that the OT cost and map can be estimated efficiently with projected SGD and minimax optimal convergence rate is given. All reviewers agree that the semi-discrete OT problem is interesting and important, and the theory and rate provided are both strong and novel. As such, the paper contribution would be a worthy addition to the literature in OT.